# Vibrio cholerae senses human enteric α-defensin 5 through a CarSR two-component system to promote bacterial pathogenicity

Yutao Liu [1,4], Tingting Xu [2,4], Qian Wang[1], Junxi Huang[2], Yangfei Zhu[3], Xingmei Liu[1], Ruiying Liu[1], Bin Yang [1✉] & Kai Zhou [2✉]

Vibrio cholerae (V. cholerae) is an aquatic bacterium responsible for acute and fatal cholera outbreaks worldwide. When V. cholerae is ingested, the bacteria colonize the epithelium of the small intestine and stimulate the Paneth cells to produce large amounts of cationic antimicrobial peptides (CAMPs). Human defensin 5 (HD-5) is the most abundant CAMPs in the small intestine. However, the role of the V. cholerae response to HD-5 remains unclear. Here we show that HD-5 significantly upregulates virulence gene expression. Moreover, a two-component system, CarSR (or RstAB), is essential for V. cholerae virulence gene expression in the presence of HD-5. Finally, phosphorylated CarR can directly bind to the promoter region of TcpP, activating transcription of tcpP, which in turn activates downstream virulence genes to promote V. cholerae colonization. In conclusion, this study reveals a virulence-regulating pathway, in which the CarSR two-component regulatory system senses HD-5 to activate virulence genes expression in V. cholerae.

[1] TEDA Institute of Biological Sciences and Biotechnology, TEDA, Nankai University, Tianjin, PR China. [2] The Second Clinical Medical College, Jinan University, The First Affiliated Hospital, Southern University of Science and Technology, Shenzhen People's Hospital, Shenzhen Institute of Respiratory Diseases, Shenzhen, Guangdong, PR China. [3] The Second Clinical Medical College, Jinan University, The First Affiliated Hospital, Southern University of Science and Technology, Laboratory Department, Shenzhen People's Hospital, Shenzhen, Guangdong, PR China. [4] These authors contributed equally: Yutao Liu, Tingting Xu. ✉email: yangbin@nankai.edu.cn; Kai_Zhou@zju.edu.cn

**V**ibrio cholerae (V. cholerae) naturally inhabits aquatic environments (e.g., rivers, estuaries, and oceans). When people ingest water or food contaminated with V. cholerae, the bacteria pass through the stomach and colonize the small intestine, causing severe watery diarrhea characteristic of cholera disease[1]. This potentially fatal diarrheal disease affects millions of people each year[2]. The gram-negative bacterium produces a variety of virulence factors and promotes colonization of the small intestine. Toxin coregulated pilus (TCP) and cholera toxin (CT) are the most essential virulence factors for colonization. ToxT, which is an AraC/XylS-like regulatory protein, directly activates the transcription of most V. cholerae virulence genes, including cholera toxin genes (ctxAB) and TCP operons[3]. The transcription of toxT is regulated by two integral membrane regulatory proteins, ToxR and TcpP[4,5]. The latter is encoded on the large pathogenicity island (called the VPI) and binds a conserved repeat element (TGTAA-N6-TGTAA) in the promoter region of toxT to activate the downstream virulence cascade[6].

As V. cholerae moves from the aquatic environment to the human small intestine, it must respond to a variety of intestinal signals, such as bile salts[7], osmolarity[8], oxygen[9], mucus[10], and pH[11], to enhance virulence gene expression and to promote colonization. Like most pathogens, V. cholerae utilizes two-component signal transduction systems (TCSs) to extensively adapt to different intestinal stimuli[12]. Previous studies have demonstrated the involvement of TCSs in virulence factor production and intestinal colonization[13–15]. In addition, CarSR or VprAB in V. cholerae is involved in virulence regulation in various pathogenic bacteria[16–22]. In V. cholerae C6706, CarR was confirmed to contribute to intestinal colonization through endotoxin modification[23,24]. However, the influence of CarSR on virulence in V. cholerae is not completely understood.

The human gastrointestinal tract has many barriers that prevent colonization by pathogenic microorganisms[25]. Cationic antimicrobial peptides (CAMPs) produced by epithelial cells contribute to defense against bacterial infections[26]. V. cholerae and other pathogens must counter CAMPs to survive[27]. CAMPs include defensins (e.g., α-defensins, β-defensins and cathelicidin LL-37), whose expression can be constitutive or activated by microorganisms[28]. Among CAMPs, human defensin 5 (HD-5) is the most abundant product secreted by Paneth cells in the small intestine and plays a crucial role in the innate immune system[29]. HD-5 kills bacteria by forming a hole in the bacterial membrane[30].

However, some pathogens, such as Shigella spp., sense HD-5 to enhance their adhesion and invasion of mucosal tissues[31]. During infection, V. cholerae moves to the intestinal crypts, where CAMPs are present, and uses two main mechanisms to resist these CAMPs. First, it confers resistance to antimicrobial peptides by upregulating expression of the almEFG operon through aminoacylation of lipopolysaccharide (LPS) via the CarSR TCS[23]. Second, when V. cholerae is incubated with CAMPs, it produces outer membrane vesicles (OMVs) containing high levels of the Bap1 protein, which traps LL-37[32]. It was shown that the expression of carR was increased by sensing polymyxin B to enhance antimicrobial peptide resistance and lipid A modification[23,24]. However, the role of HD-5 in V. cholerae pathogenesis remains unclear.

In this study, we aimed to study virulence regulatory pathway through which V. cholerae senses HD-5 via CarSR to enhance virulence gene expression and pathogenicity. CarR activated by HD-5 directly was found to induce the expression of tcpP and activates downstream virulence genes. Finally, CarSR could sense other CAMPs, such as LL-37, HBD2 and HD-6, for successful colonization in vivo.

## Results

**Transcriptome analysis of V. cholerae in response to HD-5.** To characterize the transcriptional regulation of HD-5 by V. cholerae, whole-genome expression profiles of wild-type (WT) V. cholerae grown with or without HD-5 (50 μg/mL) were compared using high-throughput Illumina RNA-seq analysis. The results indicated that 123 genes were differentially expressed, with 41 downregulated and 82 upregulated (fold change <0.5 and >2; $P \leq 0.05$) (Fig. 1a, Supplementary Data 1). Ten of the differentially expressed genes were randomly selected for qRT-PCR to validate the RNA-seq data. Our results showed that the expression trends of the ten genes were consistent between RNA-seq and qRT-PCR data, suggesting that RNA-seq were reliable in identifying transcriptional changes (Fig. 1b).

Genes displaying differential expression in the presence and absence of HD-5 were classified using the NCBI Clusters of Orthologous Groups (COG) functional categories annotation system. The COG categories that were significantly enriched in the set of upregulated genes were primarily involved in defense mechanisms, intracellular trafficking, secretion and vesicular transport, signal transduction mechanisms, amino acid transport

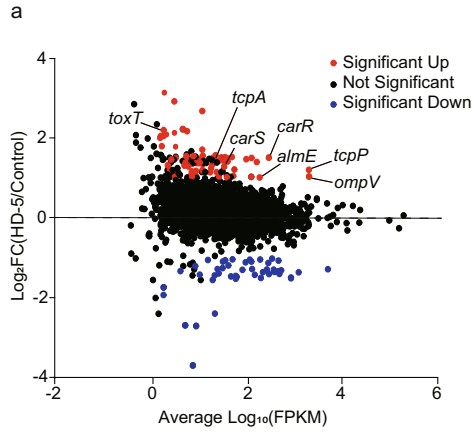
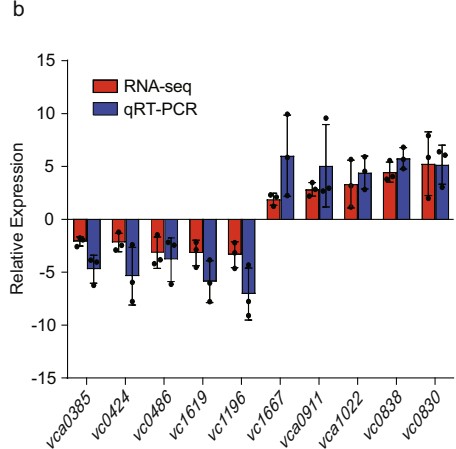

**Fig. 1 Transcriptome analysis of V. cholerae in response to HD-5. a** MA plot of differential gene expression levels in V. cholerae EL2382 in AKI medium containing 0 and 50 μg/mL HD-5. Each dot represents a single gene. Red dots represent genes whose abundance is significantly upregulated, blue dots downregulated, and black dots unchanged (or non-significantly changed) regulation. **b** RNA-seq results validation by qRT-PCR. Data are presented as mean ± SD ($n = 3$).

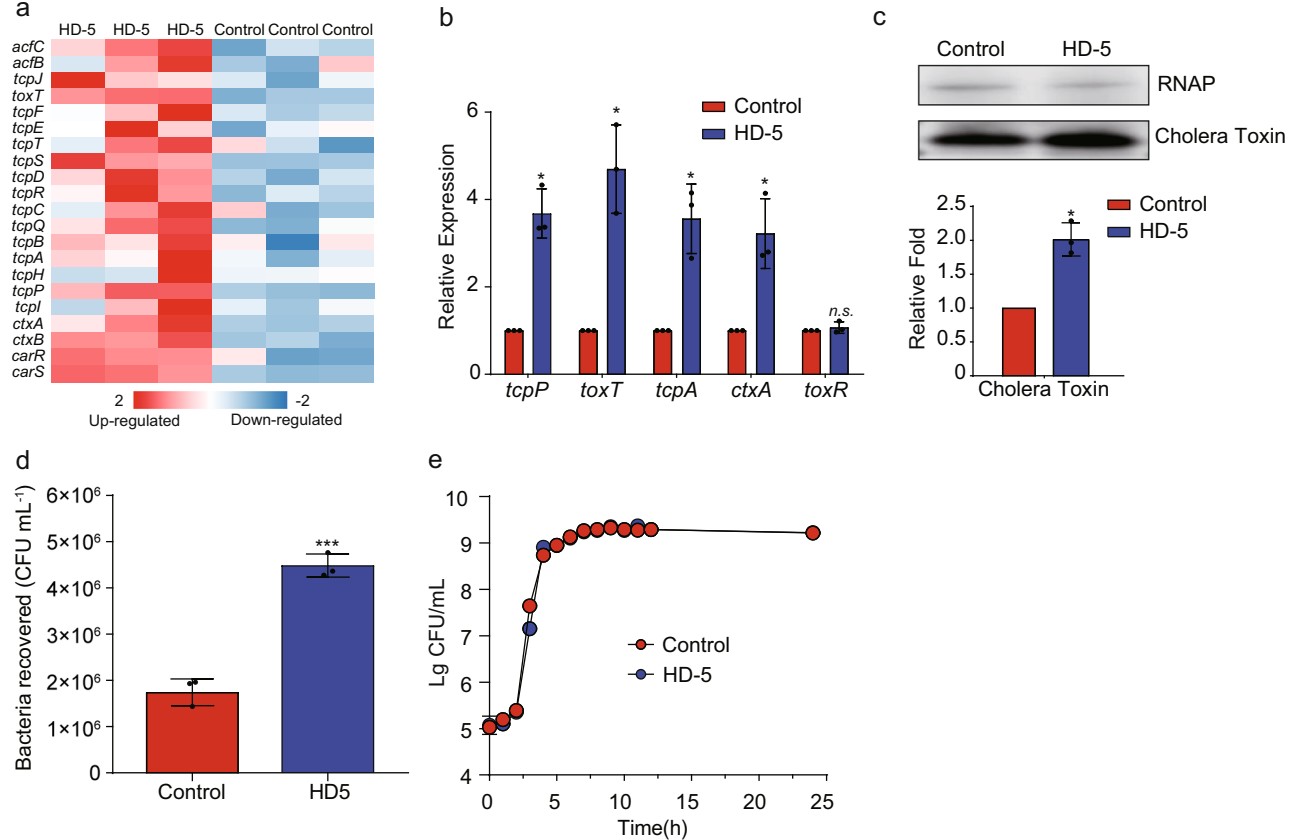

**Fig. 2 HD-5 influences *V. cholerae* colonization and virulence gene expression. a** Heatmap of differentially expressed genes in AKI medium containing 0 and 50 µg/mL HD-5. The z score indicates whether the genes were upregulated (red) or downregulated (blue). **b** qRT-PCR analysis of virulence genes expression in AKI medium containing 0 (Control) and 50 µg/mL HD-5 (HD-5). Data are presented as mean ± SD (*n* = 3). **c** Western blotting and quantitative analysis of cholera toxin in AKI medium containing 0 and 50 µg/mL HD-5. RNA polymerase (RNAP) was used as a control. Data are presented as mean ± SD (*n* = 3). **d** Adherence of *V. cholerae* to Caco-2 cells containing 0 and 50 µg/mL HD-5. Data are presented as mean ± SD (*n* = 3). **e** Growth of *V. cholerae* in AKI medium containing 0 and 50 µg/mL HD-5. Data represents the mean ± SD (*n* = 3). Significance was determined by two-tailed unpaired Student's *t* test (**b–d**). *$P \leq 0.05$; **$P \leq 0.01$; ***$P \leq 0.001$; n.s. no significant difference.

and metabolism, lipid transport and metabolism, transcription, replication, recombination and repair, cell wall/membrane/ envelope biogenesis, posttranslational modification, protein turnover, chaperones and inorganic ion transport, and metabolism (Supplementary Fig. 1a). The COG categories that were significantly enriched in the set of downregulated genes included translation, ribosomal structure and biogenesis, protein turnover, and nucleotide transport and metabolism (Supplementary Fig. 1a). Genes with unknown functions were not shown.

**HD-5 influences *V. cholerae* colonization and virulence gene expression**. Based on RNA-seq results, we discovered that the expression of most virulence genes was significantly upregulated in the presence of HD-5 (Figs. 1a, 2a). These results demonstrate that HD-5 may activate the virulence of *V. cholerae*. To verify whether HD-5 is related to virulence gene expression in *V. cholerae*, five key virulence genes (*toxT, tcpP, tcpA, ctxA,* and *toxR*) were detected by qRT-PCR. Transcript levels of *toxT, tcpP, tcpA* and *ctxA* were significantly upregulated in *V. cholerae* treated with 50 µg/mL HD-5 compared with those in the untreated control (0 µg/mL HD-5) (Fig. 2b). However, *toxR* expression was not significantly changed, indicating that HD-5 induced TCP operon upregulation is not mediated by *toxR*. Western blotting showed that the production of cholera toxin was significantly higher in the presence of HD-5 than in the control in AKI medium (Fig. 2c). Moreover, cell adherence assays revealed that

*V. cholerae* significantly enhanced the adherence capacity to Caco-2 cells in the presence of 50 µg/mL HD-5 (Fig. 2d). Importantly, bacterial growth remained unaffected under these conditions (Fig. 2e), which suggests that HD-5-dependent virulence regulation is not due to a metabolic defect caused by the absence of HD-5. Collectively, these results suggest that *V. cholerae* may sense HD-5 to promote expression of virulence-associated genes.

**CarR is activated in response to HD-5**. To elucidate the mechanism of *V. cholerae* virulence regulation mediated by HD5, the RNA-seq results were further analyzed. We observed that the expression of a TCS, called *carSR* (*vc1319* and *vc1320*), was significantly upregulated in the presence of HD-5 (Fig. 2a). qRT-PCR analysis showed a 3.92- and 2.53-fold increase in the expression of *carR* and *casS* in AKI medium supplemented with HD-5 (Fig. 3a). In addition, qRT-PCR assays showed that the expression of *carR* and *casS* in WT strain colonized in the small intestine exhibited a 10.18- and 3.51-fold increase compared to that in LB medium (Fig. 3b). Western blotting showed that CarR protein level exhibited a significant increase in AKI medium supplemented with 50 µg/mL HD-5 (Fig. 3c). Furthermore, the phosphorylation level of CarR in AKI medium supplemented with 50 µg/mL HD-5 was significantly increased compared to that with 0 µg/mL HD-5 (Fig. 3d). These results indicated that HD-5

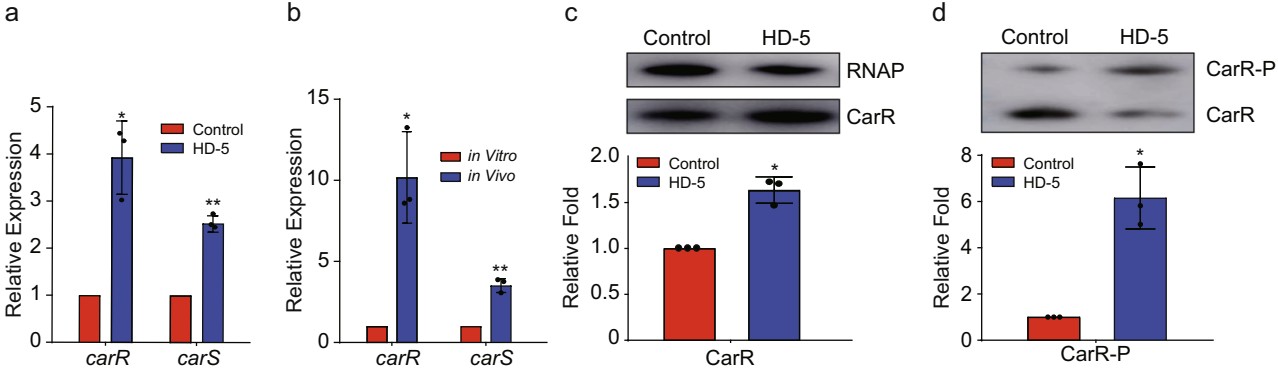

**Fig. 3 CarR is activated in response to HD-5. a**, **b** qRT-PCR analysis of *carR* and *carS* expression in AKI medium containing 0 and 50 μg/mL HD-5 (**a**) and in the small intestine of mice (**b**). Data are presented as mean ± SD (*n* = 3). **c** Western blotting and quantitative analysis of CarR in AKI medium containing 0 and 50 μg/mL HD-5. RNA polymerase (RNAP) was used as a control. Data are presented as mean ± SD (*n* = 3). **d** Phosphorylation status analysis of CarR in AKI medium containing 0 and 50 μg/mL HD-5. CarR-P phosphorylated CarR, CarR non-phosphorylated CarR. Significance was determined by a two-tailed unpaired Student's *t* test (**a**–**d**). *$P \leq 0.05$; **$P \leq 0.01$; ***$P \leq 0.001$; n.s. no significant difference.

promotes *carR* expression, CarR protein level and phosphorylation activity.

**CarR is required for *V. cholerae* pathogenicity.** To investigate whether CarR regulates the pathogenicity of *V. cholerae*, a *carR* mutant (Δ*carR*) and *carR* complement strain (Δ*carR+*) were constructed. An in vivo competition assay showed that the CI values of Δ*carR* and Δ*carR+* versus WT strain in the luminal content were 1.09 and 1.22, respectively, while the CI values in the epithelium were 0.38 and 1.19. These results indicate that CarR promotes *V. cholerae* colonization in the mouse small intestine by increasing adhesion to the intestinal epithelium, but not by increasing bacterial growth in the intestinal lumen (Fig. 4a). In addition, qRT-PCR results showed that Δ*carR* decreased the expression of virulence genes (*tcpP*, *toxT*, *tcpA*, and *ctxA*) and that their expression can be restored upon complementation in vitro and in the small intestine (Fig. 4b, c). Moreover, virulence gene expression was not induced in Δ*carR* after HD-5 treatment (Fig. 4b), suggesting that HD-5 induces virulence gene expression via CarR. Western blot results showed that cholera toxin production in Δ*carR* was significantly lower than that in WT and was restored by complementation of Δ*carR* (Fig. 4d). Bacterial growth curves in AKI medium supplemented with 50 μg/mL HD-5 showed that Δ*carR* exhibited a lower growth rate and reached a lower final cell counts compared to WT (Fig. 4e). This result suggests that CarR enhances resistance to HD-5, thereby maintaining the normal growth of *V. cholerae*. Therefore, CarR regulates both the virulence and HD-5 resistance in *V. cholerae*.

**CarR modulates virulence gene expression by directly regulating *TcpP*.** To further investigate the molecular aspects of CarR regulation, we performed a ChIP-seq assay. The resulting data included 49 peaks, from which 5 (fold enrichment > 1.4) were randomly selected for validation by ChIP-qPCR (Supplementary Data 2). ChIP-qPCR results confirmed that P$_{vc1318}$, P$_{vca1078}$, P$_{vc0973}$, P$_{vc0633}$, and P$_{vca0227}$ were enriched in CarR-ChIP samples (Fig. 5a), showing that the ChIP-seq data are reliable. Further analysis of ChIP-seq data to reveal potential direct targets of CarR regulation was performed. Among these CarR-binding targets, one peak was within the promoter region of *vc0826* (*tcpP*), which is a positive regulator of virulence gene expression in *V. cholerae*. Therefore, we hypothesized that activation of virulence gene expression by CarR is mediated by TcpP. ChIP-

qPCR confirmed the binding of CarR to the *tcpP* promoter (P$_{tcpP}$) in vivo. The results showed that P$_{tcpP}$ and P$_{almE}$ (positive control) were 3.24- and 11.2-fold enriched in CarR-ChIP samples compared to those in the mock-ChIP samples (Fig. 5b). Conversely, fold enrichments of P$_{toxT}$ and coding region of *rpoS* (negative control) did not differ between the CarR-ChIP and mock-ChIP samples (Fig. 5b). Next, we evaluated the binding of CarR to P$_{tcpP}$ via electrophoretic mobility shift assays (EMSAs). As shown in Fig. 5c, at increasing concentrations of phosphorylated CarR protein, slow migrating bands were observed for the FAM-labeled promoters of *tcpP*. Moreover, addition of unlabeled promoters effectively competed for CarR binding to the labeled promoters, and the retarded band disappeared in the presence of 100-fold excess unlabeled promoter DNA (Fig. 5c and Supplementary Fig. 2a). These results indicate that phosphorylated CarR binds specifically to the promoter regions of *tcpP* and *almE* in vitro. In addition, the binding capacity of non-phosphorylated CarR to the promoter regions of *tcpP* and *almE* was significantly reduced compared with that of phosphorylated CarR (Fig. 5d and Supplementary Fig. 2b). Meanwhile, phosphorylated CarR did not bind to the negative control (the promoter region of *toxT* and coding regions of *rpoS*) under the same experimental conditions (Supplementary Fig. 2c, d). These results confirm that phosphorylated CarR directly binds to P$_{tcpP}$.

To identify the CarR-binding site in the promoter region of *tcpP*, we performed a dye-based DNase I footprinting assay and found that CarR binds to a specific sequence containing a 45-bp motif, which is located −198 to −152 from the translational start site (Fig. 5e). Further sequence analysis showed that a potential RstA box in *E. coli* (5-TAATGAGAATTACT-3) from −186 to −172 bp was found[33,34]. To further determine whether the 45-bp motif and the potential RstA box are important for binding to CarR, we performed EMSA and competition assays again using a P$_{tcpP}$-1 DNA fragment (without the 45-bp motif) and a P$_{tcpP}$-2 DNA fragment (without the potential RstA box) under the same conditions (Fig. 5f, g). Neither the P$_{tcpP}$-1 nor P$_{tcpP}$-2 DNA fragment was able to bind to CarR, confirming that the potential RstA box is crucial to the binding ability of CarR to P$_{tcpP}$.

Given that CarR regulates *tcpP* expression, we wanted to determine if the fitness defect of the Δ*carR* observed in Fig. 4a was due to this effect alone. We therefore constructed a Δ*tcpP* and a Δ*carR*Δ*tcpP* double mutant in *V. cholerae* and competed them against a Δ*carR lacZ−* alone. The colonization assay showed the CI value of Δ*carR*Δ*tcpP* versus Δ*carR lacZ-* was 1.09 and 0.285 in luminal contents and the tissue-associated contents of mice,

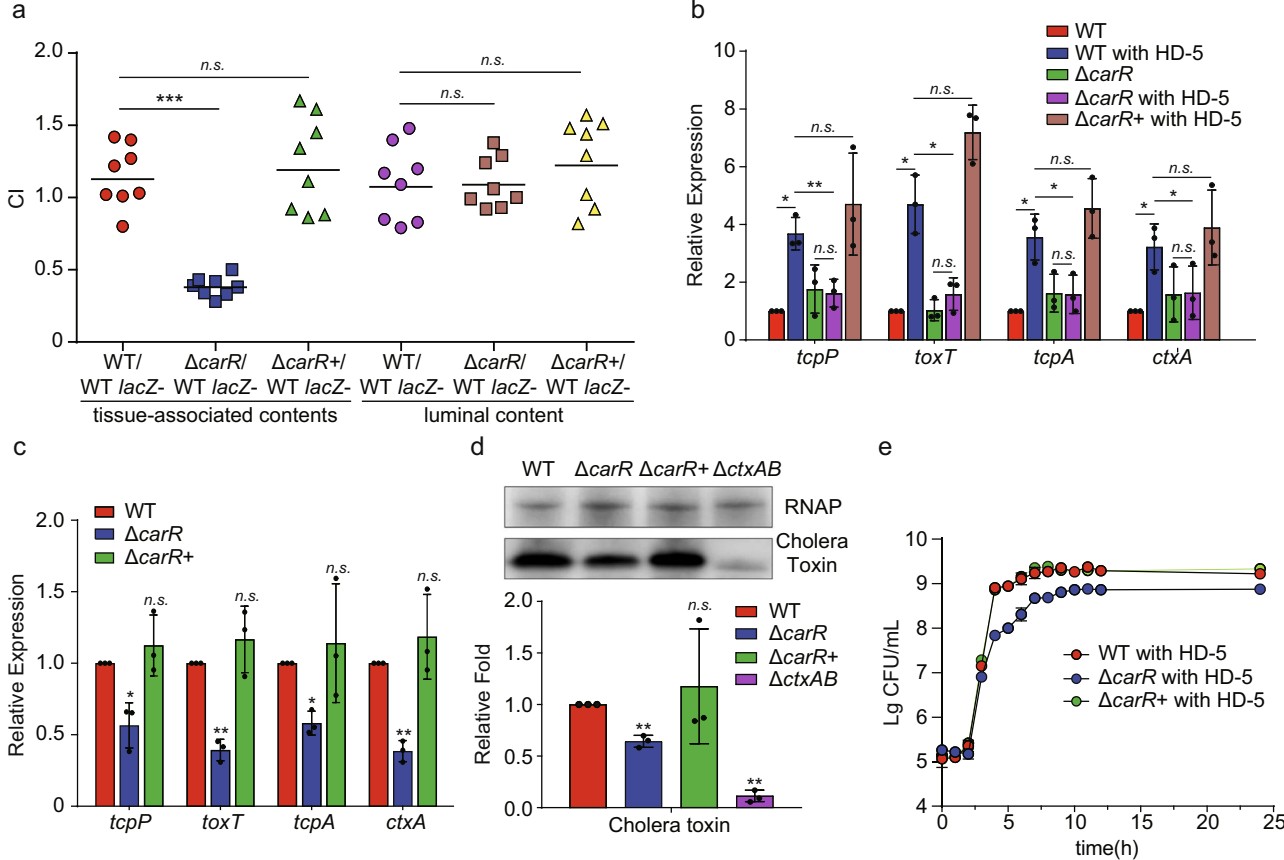

**Fig. 4 CarSR is required for *V. cholerae* pathogenicity. a** In vivo competition assay of WT, Δ*carR*, and Δ*carR*+ in luminal contents and tissue-associated contents (*n* = 8). CI is defined as the output ratio of mutant strains to WT *lacZ*− divided by the input ratio of mutant strains to WT *lacZ*−. Each symbol represents the CI in an individual mouse; horizontal bars indicate the median. **b, c** qRT-PCR analysis of virulence genes expression in WT, Δ*carR*, and Δ*carR* + in AKI medium containing 0 and 50 μg/mL HD-5 (**b**) and in the small intestine of mice (**c**). Data are presented as mean ± SD (*n* = 3). **d** Western blotting and quantitative analysis of cholera toxin in WT, Δ*carR*, Δ*carR*+ and Δ*ctxAB* in AKI medium containing 50 μg/mL HD-5. RNA polymerase (RNAP) was used as a loading control. Δ*ctxAB* was used as a negative control. Data are presented as the mean ± SD (*n* = 3). **e** Growth of WT, Δ*carR*, Δ*carR*+ in AKI medium containing 50 μg/mL HD-5. Data represents the mean ± SD (*n* = 3). Significance was determined by two-sided Mann–Whitney *U* test (**a**) or two-tailed unpaired Student's *t* test (**b–d**). *$P \leq 0.05$; **$P \leq 0.01$; ***$P \leq 0.001$; n.s. no significant difference.

which was similar to the CI value (1.22 and 0.327) of Δ*tcpP* versus Δ*carR lacZ*− in the luminal contents and the tissue-associated contents (Fig. 5h). This suggests that the *carR* deletion did not affect *V. cholerae* colonization in the Δ*tcpP* background. Together, these results indicate that the influence of TcpP on the colonization of *V. cholerae* is mediated by CarR.

**CAMPs promote *V. cholerae* adhesion and virulence genes expression through similar molecular mechanisms.** CAMPs are widespread in the small intestine and include LL-37, human β-defensins, HD-5, and HD-6[35–37]. CAMPs disrupt the bacterial cell envelope by binding to the anionic lipid A membrane anchor of major cell surface molecules, lipopolysaccharide (LPS), and acid glycerophospholipids, except for HD-6[38]. HD6 lacks the broad-spectrum antimicrobial activity observed for other human α-defensins[39]. Previous studies have shown that CarR promotes polymyxin B resistance by upregulating the expression of *almEFG*. The AlmEFG proteins can modify lipid A by the addition of glycine and diglycine[23]. Moreover, a TCS called PmrAB regulates resistance to several CAMPs, including polymyxins B and E, cattle indolicidin, and LL-37, in *Pseudomonas aeruginosa*[40].

We speculated that CAMPs, including LL-37, HBD2, and HD6, may regulate *V. cholerae* pathogenicity *via* a similar mechanism.

qRT-PCR results showed that the expression of *carR* and *almE* exhibited a 4.33- to 15.87-fold and 2.57- to 27.86-fold, respectively, increased expression to LL-37, HBD2, and HD-6 (Fig. 6a). These results indicate that different CAMPs can active expression of *carSR* and downstream genes in vitro. To further investigate whether CAMPs activate pathogenicity *via* CarR, qRT-PCR and adherence assays were performed. The expression of virulence genes was upregulated in WT *V. cholerae* but showed no difference in Δ*carR* when containing CAMPs (Fig. 6b). Consistent with these results, the adherence assay showed that in the presence of CAMPs, the adherence capacity significantly increased in WT *V. cholerae* (Fig. 6c). Conversely, adherence capacity was not induced in Δ*carR* in the presence of CAMPs (Fig. 6d). These results indicate that CAMPs induce adherence and that the response is mediated by CarSR in vitro.

Adherence to intestinal epithelial cells can be affected by autoaggregation[41,42], therefore we tested the autoaggregation of the WT and Δ*carR*. The result showed that the A600 values of both WT and Δ*carR* decreased substantially after 1 h incubation at 37 °C in DMEM media (Supplementary Fig. 3a). This data confirmed that *carR* regulates adherence capacity via modulating virulence gene expression, rather than by regulating autoaggregation capacity. Taken together, these results suggest that CAMPs promote *V. cholerae* adhesion and virulence gene expression through similar molecular mechanisms.

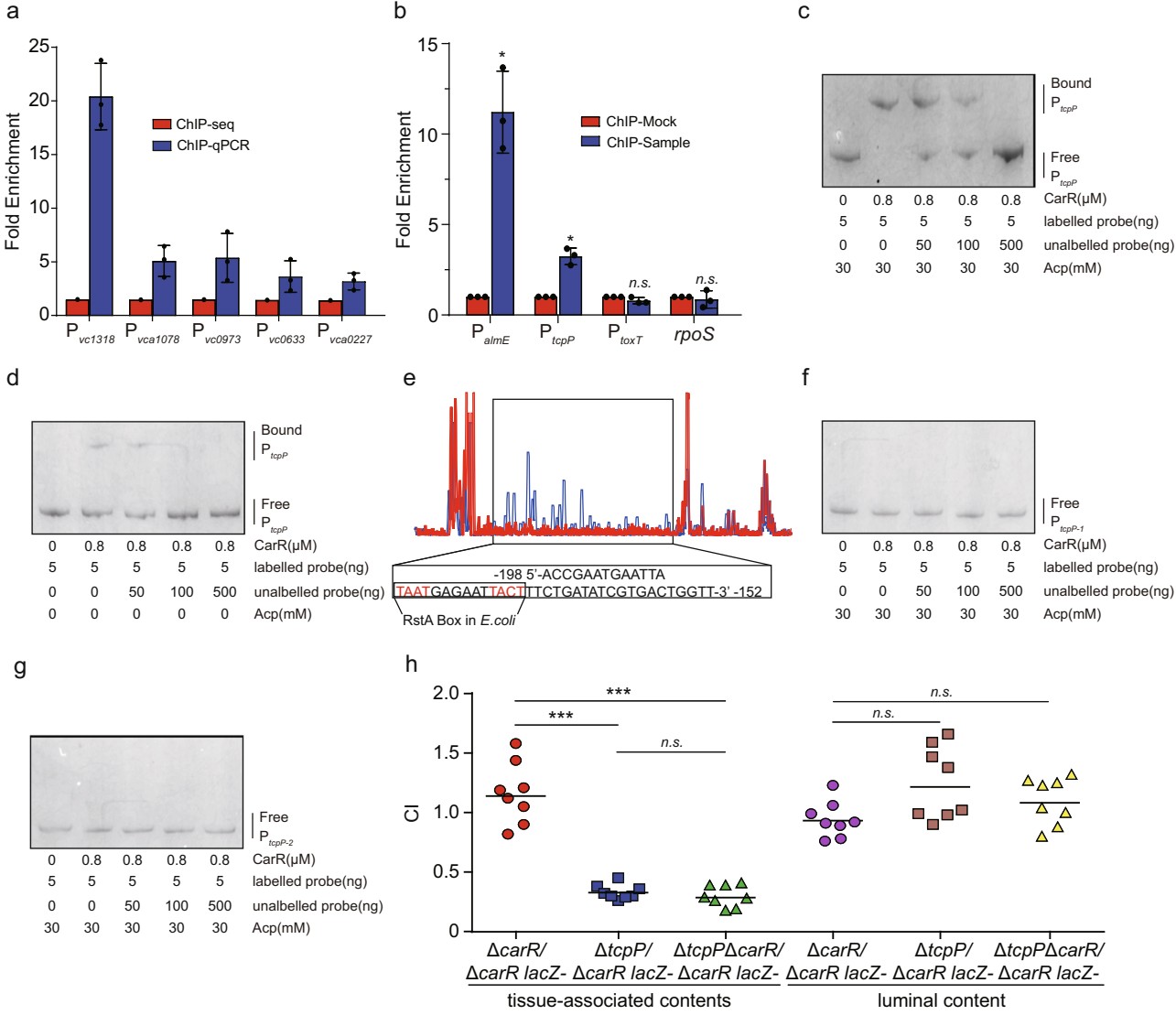

**Fig. 5 CarR modulates virulence gene expression by directly regulating TcpP. a** ChIP-seq results validation by ChIP-qPCR. Fold enrichment of P$_{vc1318}$, P$_{vca1078}$, P$_{vc0973}$, P$_{vc0633}$, and P$_{vca0227}$ in CarR-ChIP samples, as measured by qPCR. **b** Fold enrichment of P$_{almE}$, P$_{tcpP}$, and P$_{toxT}$ in CarR-ChIP samples, as measured by qPCR. P$_{almE}$ served as a positive control, and *rpoS* and P$_{toxT}$ served as negative controls. Data are presented as mean ± SD ($n = 3$). **c** CarR binds to a motif in the *tcpP* promoter region. The protected region shows a significantly reduced peak intensity (red) pattern compared with those of control (blue). The identified RstA box in *E. coli* is shown in a box at the bottom of the figure. **d**, **e** CarR bound to P$_{tcpP}$ with (**d**) or without (**e**) 30 mM acetyl phosphate. **f**, **g** CarR bound to 45-bp motif deletion (**f**) and potential RstA box deletion (**g**) P$_{tcpP}$ with 30 mM Acetyl phosphate. **h** In vivo competition assay of Δ*carR*, Δ*tcpP*, and Δ*tcpP*Δ*carR* in luminal contents and tissue-associated contents ($n = 8$). CI is defined as the output ratio of Δ*tcpP* and Δ*tcpP*Δ*carR* to Δ*carR lacZ-* divided by the input ratio of Δ*tcpP* and Δ*tcpP*Δ*carR* to Δ*carR lacZ−*. Each symbol represents the CI in an individual mouse; horizontal bars indicate the median. Significance was determined by a two-sided Mann–Whitney *U* test (**h**) or two-tailed unpaired Student's *t* test (**b**). *$P \leq 0.05$; **$P \leq 0.01$; ***$P \leq 0.001$; n.s. no significant difference.

## Discussion

In this study, we elucidated the molecular mechanisms used by *V. cholerae* to sense CAMPs and confer a colonization advantage in the small intestine. We have now provided a model to link nuances of transcriptional regulation of TCP, quorum sensing, small RNA regulation and antimicrobial resistance regulatory pathway (Fig. 7). Briefly, when *V. cholerae* enters the small intestine at the initial stage of colonization, quorum sensing (QS) is not induced. Four small RNAs called Qrr1-4 activate translation of AphA and inhibit production of HapR. Meanwhile, CAMPs in small intestine interact with the lipopolysaccharide of *V. cholerae* and gain access to the periplasmic space[43,44]. CarS is activated by periplasmic CAMPs and activates the phosphorylated state of CarR. Consequently, expression of *tcpP* and *almE*

are upregulated, thereby promoting bacterial colonization and antimicrobial resistance. During later stages of infection, as population density increases, the increased cellular level of cyclic adenosine monophosphate (cAMP) induces an efflux of water into the lumen of the small intestine that causes watery diarrhea. At this stage, CarS is inactivated due to the decreased CAMP concentration and HapR is induced at high cell density. During these processes, *V. cholerae* down-regulates the expression of virulence factors and detaches to disseminate throughout the small intestine.

Although CAMPs may differ in length and amino acid composition, most CAMPs share common structural characteristics and similar mechanisms of action. The mechanism of CAMP action on bacteria is mainly realized through electrostatic

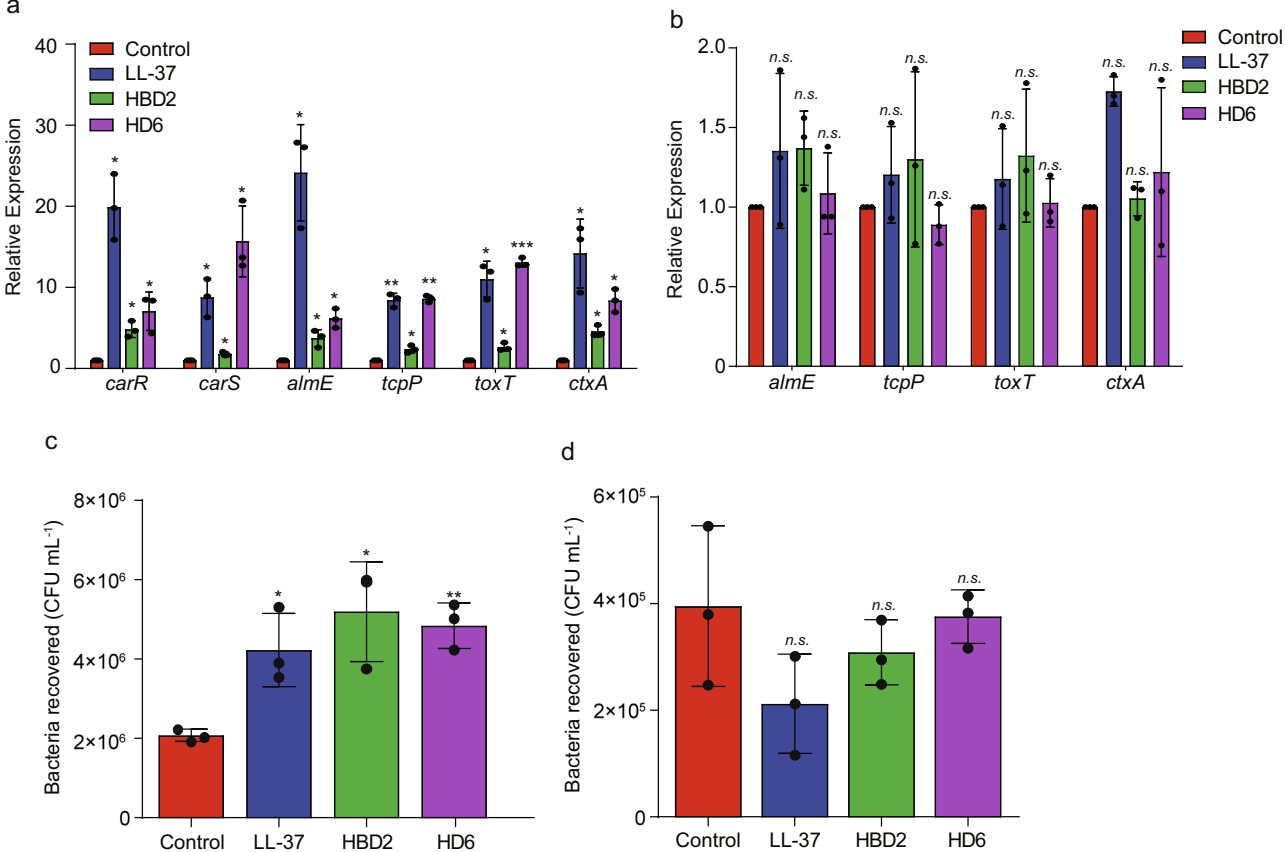

**Fig. 6 CAMPs promote V. cholerae adhesion and virulence gene expression through similar molecular mechanisms. a, b** qRT-PCR analysis of the expression of corresponding genes in WT (**a**) and ΔcarR (**b**) in AKI medium containing 50 μg/mL LL-37, HBD2, and HD-6. Data are presented as the mean ± SD ($n = 3$). **c, d** Adherence of WT (**c**) and ΔcarR (**d**) to Caco-2 cells containing 50 μg/mL LL-37, HBD2 and HD-6. Data are presented as mean ± SD ($n = 3$). Significance was determined by two-tailed unpaired Student's t test (**a–d**). *$P ≤ 0.05$; **$P ≤ 0.01$; ***$P ≤ 0.001$; n.s. no significant difference.

membrane interactions. In this study, we showed that similar to the mechanism of *V. cholerae* against HD-5, other CAMPs (e.g., LL-37, HBD2, and HD-6) also promoted adherence and virulence gene expression through CarSR during infection. HD-5 is widespread in the small intestine and is an important defense against invasion by pathogenic microorganisms. However, an increasing number of reports have described bacteria as resistant to HD-5. For instance, Shi*gella* spp. regulate innate immunity by sensing HD-5 as a virulence factor for infection[31]. In *V. cholerae*, CarSR was identified to sense polymyxin B and directly regulate the expression of *almEFG* to resist killing by polymyxin B. However, the Δ*almEFG* did not exhibit defects in the colonization capacity of *V. cholerae* and the contribution of CarR to colonization differs between the *V. cholerae* strains[23,24]. In this study, we revealed that, in *V. cholera* strain EL2382, CarSR promotes virulence gene expression, adhesion in vitro and intestinal colonization in vivo. So, we speculated that *V. cholerae* sense CAMPs in small intestine to activate CarSR, which would further promote the expression of *tcpP* and downstream virulence genes to confer colonization advantages. Given that CarSR regulation pathway is strain specific, whether this phenomenon also exists in other *V. cholerae* strains need further investigation.

Based on the RNA-seq results, some other interesting genes appeared to be upregulated. The products of these genes are mainly involved in including lipid A modification (*vc1579*), LPS biosynthesis (*vc0235*), oligopeptide ABC transporter (*vc1092*, *vc1093*), amino acid ABC transporter (*vc1359*; *vc1863*; *vc1864*), long-chain fatty acid transport (*vca0862*), putative adhesion (*vc1318*) and transcription regulation (*vc1741*, *vca0642* and

*vca0704*). These results suggest that *V. cholerae* greatly promotes the stress defense capacity of *V. cholerae* against HD5 by sensing HD-5 and regulating the expression of multiple genes simultaneously. For instance, *V. cholerae* may promote the expression of genes encoding ABC transporters to pump HD-5 out of the bacterial cells or modify LPS-lipid A to confer HD-5 resistance. Previous research showed the expression profiles of ΔcarR to WT grown in LB media. So, a comparison was made between previous Δ*carR* RNA-seq results with our HD-5 RNA-seq results. These results showed that *vc1579* (*almE*) and *vc1318* (*ompV*) were significantly downregulated in Δ*carR* but significantly upregulated in HD-5 RNA-seq results. *almE* has been proved positively regulated by *carR* to promote polymyxin B resistance. OmpV can help in adhesion and invasion of intestinal epithelial cell by *Salmonella*[45,46]. However, the role of *ompV* in *V. cholerae* needs further investigation.

The mammalian gut is a complex environment. To successfully colonize the gut, pathogens must compete with symbiotic bacteria for nutrients and an ecological niche, sense changes in environmental signals, and precisely regulate the expression of virulence genes. Bacteria sense and respond to environmental changes through TCSs. Multiple TCSs are implicated in bacterial pathogenicity. CarR is a TCS regulator that regulate virulence in different bacterial species. In this study, we found that phosphorylated state of CarR directly binds to P*tcpP*. This suggests phosphorylated CarR in *V. cholerae* directly activates the expression of the *tcpP* gene and induces the expression of downstream virulence genes. Furthermore, a specific CarR-bound sequence (5-TAATGAGAATTACT-3; −186 to −172 from the

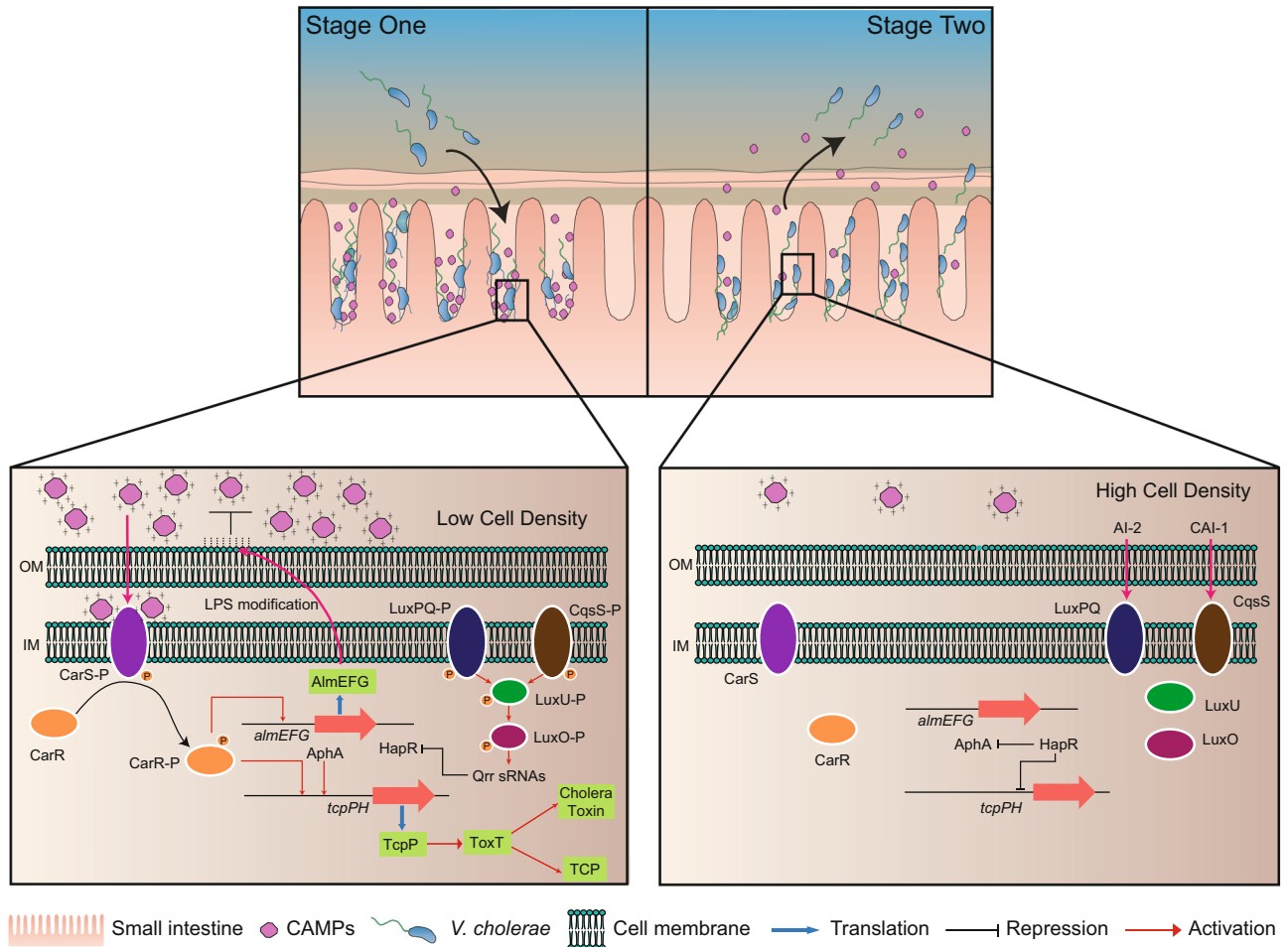

**Fig. 7 Graphic model of CarSR-mediated HD-5 signaling pathway in *V. cholerae*.** At initial stage (Stage One) of colonization, CarSR is activated by sensing HD-5 to promote the expression of *tcpP* and *almE*. At late stage (Stage Two) of infection, CarSR is inactivated due to the decreased HD-5 concentration, which lead to the repression of virulence genes expression.

*tcpP* translational start site) was confirmed. Previous research showed that several regulators could directly bind to the *tcpP* promoter, including AphA, AphB, HapR and CRP[47,48]. A small RNA (~91 nt), which was named TarA, is also located in the *tcpP* promoter[49]. A binding site analysis revealed that CarR-binding site for *tcpP* promoter overlaps with neither the binding sequence of AphA, AphB, HapR and CRP nor the coding sequence of *tarA* (Supplementary Fig. 4a). This indicated that CarR and other regulators cooperate or compete to regulate *tcpP* expression. In addition, CarR was found to bind several other promoters of genes. These genes include the genes encoding putative hemolysin (*vc0489, vca0218, vca0219*), encoding regulators (*vc1021; vc1617; vc1741; vca1078*), encoding adhesin factors (*vc1318; vc0633*), encoding transporters (*vc0069; vc1325; vc1669; vca0205; vca0227*), encoding amino acid ligase (*vc1579*) and encoding tryptophanase (*vca0161, tnaA*). This suggests that CarR plays multiple roles during *V. cholerae* infection. Combined analysis of RNA-seq and ChIP-seq data revealed that some other upregulated genes induced by HD-5 were also identified in ChIP-seq results. These genes include *vc1318* (*ompV*) encoding putative adhesin, *vc1579* (*almE*) encoding lipid A modification system glycine--protein ligase, *vc1727* (*glgC*) encoding glucose-1-phosphate adenylyl-transferase and *vc1741* encoding a regulator. These results indicated that *V. cholerae* CarR directly regulates the expression of genes involved in antimicrobial resistance (*almE*), adhesin (*ompV*), metabolism (*glgC*) and regulation (*vc1741*) pathways in

response to HD-5. Considering all these genes are not under the control of TcpP, thus CarR regulates the expression of these genes uniquely and independent of TcpP.

Cholera is an acute diarrheal disease that can be fatal within hours if untreated. There are an estimated 3 to 5 million cases of cholera each year, resulting in 100,000 to 120,000 deaths. In this study, we improved the understanding of how *V. cholerae* utilizes intestinal signals to facilitate intestinal colonization. These findings provide a paradigm of *V. cholerae* signal perception and virulence regulation in the small intestine. Additionally, our results can be employed to investigate other pathogens in the human gastrointestinal tract. These results highlight that host defense factors, such as CAMPs, can be important contributors to pathogenesis by *V. cholerae*, which may redefine the role as therapeutic drugs. Moreover, CarSR may be used as a therapeutic target for the prevention and treatment of cholera.

## Methods

**Bacterial strains and culture conditions.** The bacterial strains and plasmids used in this study are listed in Supplementary Table 1. The *V. cholerae* O1 El Tor strain El2382 isolated in 1994, was kindly provided by Shanghai Municipal Center for Disease Control & Prevention. *Escherichia coli* strains were grown at 37 °C in LB broth. The *V. cholerae* strains were grown overnight and diluted 1:100 in fresh AKI medium. Cultures were grown for 4 h anaerobically and reached an optical density of 0.2, followed by 2–2.5 h of shaking until reaching an optical density of 1.0. Concentrations of antibiotics (Sigma, St Louis, MO, USA) and inducers were as follows: kanamycin, 50 µg/mL; polymyxin B, 50 µg/ml; chloramphenicol, 30 µg/mL

(for *E. coli*) or 5 µg/mL (for *V. cholerae*); isopropyl β-D-1-thiogalactopyranoside (IPTG), 0.5 mM; L-arabinose, 0.2% (wt/vol); and HD-5(SP-ADF5-1, Innovagen, Sweden), HD-6 (SP-ADF6-5, Innovagen, Sweden), and HBD2 (HY-P7135, MCE, China) and LL-37 (HY-P1222, MCE, China), 50 µg/mL.

**Mutant construction, complementation, and overexpression**. The primers used in this study are listed in Supplementary Table 2. Mutants were constructed using the suicide vector pRE112[50]. The mutations were confirmed by PCR and DNA sequencing. For complementation, *carR* was amplified with its promoter regions by PCR and cloned into pBAD33. For ChIP-seq and ChIP-qPCR, *carR* was amplified with its promoter regions and cloned into pBAD33 in frame with a C-terminal 3× FLAG-tag. For CarR purification, *carR* was amplified and cloned into pET28a.

**RNA isolation, purification, and sequencing**. To study the effect of HD-5 on *V. cholerae* gene expression, the El2382 WT strain was grown overnight and diluted 1:100 in fresh AKI medium. When the cultures reached an optical density of 0.2, HD-5 was added at a concentration of 50 µg/mL and followed by 2–2.5 h of shaking until an optical density of 1.0 was reached. Total RNA was extracted by TRIzol LS Reagent (Invitrogen, Carlsbad, CA, USA) according to the manufacturer's instructions. The RNA was further purified using the RNeasy Mini Kit (Qiagen, Valencia, CA, USA). Next, the total RNA amount was determined by a NanoDrop 2000 spectrophotometer (Thermo Fisher Scientific, MA, USA). Three independent experiments were performed.

Complementary DNA (cDNA) was prepared, and sequencing was processed by Novogene, Inc. (Tianjin, China). To present gene expression levels, RPKM[51] was used as a normalization metric.

**Growth assay**. To determine the growth curve of each strain, overnight cultures were diluted 1:1000 in a flask containing 200 ml of LB broth with or without HD-5 and incubated at 37 °C with shaking at 180 rpm. A 100 µl aliquot was removed from the flask and suitable dilutions were plated on LB agar plates. The growth curve was determined by cell counts and is expressed in $\log_{10}$ CFU/ml. Experiments were independently performed three times.

**qRT-PCR**. Total RNA was extracted by using Trizol Reagent[50]. cDNA was synthesized using PrimeScript™ RT Reagent Kit with gDNA Eraser (Takara, Kusatsu, Japan) according to the manufacturer's instructions. qRT-PCR was performed in triplicate, and the results were analyzed by an Applied Biosystems ABI 7500 sequence detection system (Applied Biosystems, CA, USA) with SYBR green. The *rrsA* gene was used as a control for sample normalization. The relative difference in gene expression was calculated as a fold change using the $2^{-\Delta\Delta CT}$ formula[52]. Each experiment was calculated from three independent experiments with three technical replicates per experiment.

**Determination of CarR phosphorylation state**. Briefly, 0.2 mL of bacterial culture (OD600 of 1.0) was treated with the addition of 20 µL of 10 M NaOH followed by 1 mL of ethanol and 180 µl of 3 M sodium acetate, pH 5.2. After chilling at −80 °C for 2 h, precipitates were collected by centrifugation, rinsed with 70% ethanol, and resuspended in 100 µL of sample buffer (160 mM Tris HCl, pH 7.5, 4% SDS, 20% glycerol, and 10% 2-mercaptoethanol). To achieve a good separation of the two forms of CarR, samples were fractionated on SDS-PAGE gels containing 50 µM Phos-tag (WAKO; 304-93521) and 40 µM MnCl₂. Subsequently, gels were washed with western transfer buffer (25 mM Tris, 192 mM glycine, 20% methanol, and 0.1% SDS) containing 5 mM EDTA for 10 min, followed by a second wash with transfer buffer for 20 min.

**Western blotting**. *V. cholerae* strains were grown overnight and diluted 1:100 in fresh AKI medium. When the cultures were grown for 4 h anaerobically and reached an optical density of 0.2, followed by 2–2.5 h of shaking until reaching an optical density of 1.0. Bacterial cells were harvested, washed, and resuspended in PBS at 4 °C. The resuspension solution was sonicated for 3 min and centrifuged at 12,000 × *g* for 10 min at 4 °C to remove cell debris. Then, the supernatants were collected and quantified by the BSA method. For western blotting analysis, equal amounts of total protein (20 µg) were boiled and separated by 4–12% SDS-PAGE and transferred onto PVDF membranes (Bio-Rad) by electroblotting. Blots for RNAP (RNA polymerase) and cholera toxin were incubated in anti-RNA polymerase beta (ab191598) and anti-cholera toxin (ab123129) rabbit antibodies, respectively, at a dilution of 1:2000. Blots for CarR and anti-CarR monoclonal antibody were used (Willget Biotech Co., Ltd.). For protein detection, blots were incubated in horseradish peroxidase-linked goat anti-rabbit IgG secondary antibodies (1:5000 dilution, Sparkjade, EF0002). Then, the Sparkjade ECL plus (Sparkjade; ED0016) detection system was used for visualization. Images were acquired using an Amersham™ Imager 600 System (General Electric Company). Protein bands were quantified using ImageJ (1.8.0) software.

**Adhesion assay**. Caco-2 cell line was purchased from the Shanghai Institute of Biochemistry and Cell Biology of the Chinese Academy of Sciences (Shanghai, China). All the cell lines of the Shanghai Institute of Biochemistry and Cell Biology

are originated from ATCC. ATCC authenticates its cell lines through morphology, karyotyping, and STR analyses. Regular inspection of cell culture for coherent morphology with ATCC source images was routinely performed.

The Caco-2 cells were subcultured in a 6-well plate. Before infection, the cells were washed with prewarmed PBS three times. The medium was replaced by DMEM without antibiotics or fetal bovine serum. When necessary, CAMPs were added to a final concentration of 50 µg/mL. Overnight bacterial cultures were washed and diluted to 10⁶/mL with DMEM. Cell monolayers were infected with ~2 × 10⁶ CFU and incubated at 37 °C for 2 h in a 5% CO₂ atmosphere[53]. Following incubation, unattached bacteria were removed by washing three times with PBS. Cells were lysed by 1 ml of 0.1% sodium dodecyl sulfate. Lysate solution was gradient-diluted and plated on LB agar containing polymyxin B (50 µg/mL). The plate was cultured overnight at 37 °C, and the attachment efficiency was determined by the number of bacteria recovered on plates. At least three independent biological replicates were prepared and analyzed.

**In vivo competition assay**. Both sexes of 5-day-old CD-1 infant mice were purchased from Beijing Vital River Laboratory Animal Technology Co. Ltd. (Beijing, China). The animal research procedures were approved by the Institutional Animal Care Committee at Nankai University and Tianjin Institute of Pharmaceutical Research New Drug Evaluation Co. Ltd. (IACUC number: 2016032102), Tianjin, China. Experiments were performed under protocol no. IACUC 2016030502. Every effort was made to minimize animal suffering and to reduce the number of animals used.

Briefly, *V. cholerae* lacZ+ and lacZ− strains were grown overnight. The overnight bacteria (lacZ+) were diluted to 10⁷ CFU/mL and mixed with an equal number of bacterial (lacZ−) strains. The mixtures were orally infected into groups of eight anesthetized mice. After 24 h of incubation, the small intestine was removed, homogenized, diluted, and plated onto LB agar plates containing 5-bromo-4-chloro-3-indoyl-β-d-galactopyranoside (X-gal) and polymyxin B (50 µg/mL). The plate was cultured overnight at 37 °C to determine the recovered bacteria and to obtain the output ratios. The competitive index (CI) was determined as the output ratio of lacZ+ to lacZ− cells divided by the input ratio of lacZ+ to lacZ− cells[54].

**Chromatin immunoprecipitation (ChIP) sequencing and ChIP-qPCR**. Exponentially growing bacterial cultures of CarR-ChIP were induced with 0.2% L-arabinose (wt/vol) and 50 µg/mL HD-5 until the mid-log phase before it was treated with 1% formaldehyde for 25 min at room temperature. Crosslinking was stopped by adding 0.5 M glycine. The samples were centrifuged at 12,000 × *g* for 10 min at 4 °C and washed three times with ice-cold PBS. Next, samples were resuspended and sonicated to generate DNA fragments of 100–500 bp. Then, the samples were centrifuged at 12,000 g for 10 min at 4 °C, and the supernatant was collected for immunoprecipitation using an anti-3× FLAG antibody (Sigma, St. Louis, United States; #F1804) and protein A magnetic beads (Invitrogen, Waltham, United States; #10002D) according to the manufacturer's instructions. An aliquot with no added antibody served as the negative control (mock). Samples were incubated with 8 µL of 10 mg/mL RNaseA for 2 h at 37 °C and with 4 µL of 20 mg/mL proteinase K at 55 °C for 2 h. DNA fragments were purified using a PCR purification kit (Qiagen, Hilden, Germany; #28104). After harvesting the CarR-ChIP and mock-ChIP DNA, the next-generation sequencing library was prepared and sequenced by Novogene, Inc[55].

To measure the enrichment of CarR-binding peaks, ChIP-qPCR was performed on an ABI 7500 sequence detection system. The 16S rRNA gene (nonspecific enrichment) was used as a reference. The relative enrichment of candidate targets was calculated as a fold enrichment using the formula $2^{-\Delta\Delta CT}$. The results are reported as the average enrichment for three biological replicates.

**Electrophoretic mobility shift assays (EMSAs)**. The CarR-His₆ was expressed in *E. coli* BL21 (DE3) and purified using nickel columns (GE Healthcare; #17057501). Protein concentrations were determined using the Bradford procedure. PCR fragments encompassing the target regions were amplified with and without 6-FAM-labeled primers and gel-purified (Sparkjade; AE0101). In each case, 5 ng of each DNA probe was incubated with increasing concentrations of proteins in binding buffer (5 mM HEPES (pH 7.9), 40 mM KCl, 1 mM dithiothreitol (DTT), 0.1 mM EDTA and 5% glycerol, with or without 30 mM acetyl phosphate)[56]. For CarR gel mobility shift assays, 30 mM acetyl phosphate was added to the binding buffer for generating phosphorylated, active CarR. For competition assays, various concentrations of unlabeled DNA fragments (0 to 500 ng) were added. The reaction mixtures were incubated for 20 min at room temperature. Samples were subjected to 6% native PAGE at 90 V for 2 h in a 0.5× Tris-borate-EDTA (TBE) running buffer. Labeled fragments were visualized using Amersham Imager 600 (GE Healthcare).

**Dye primer-based DNase I footprinting assay**. A 300-bp fragment of the *tcpP* promoter regions was generated by PCR with 6-FAM primers. Forty nanograms of 6-FAM-labeled *tcpP* promoter was incubated with different amounts of CarR protein ranging from 0 to 1 µM in binding buffer (10 mM Tris-HCl [pH 7.5], 0.2 mM dithiothreitol, 5 mM MgCl₂, 10 mM KCl, and 10% glycerol). 0.05U DNase

I (Thermo; EN0521) was added to a 20-µL reaction for 5 min at room temperature. The reaction was stopped by heating at 65 °C for 10 min in the presence of 250 mM EDTA. DNA fragments were purified with the QIAquick PCR Purification kit (Qiagen; 28104) and eluted in 15 µl distilled water. Samples were analyzed by MAP Biotech CO., Ltd (Shanghai China). The results were analyzed with a peak scanner (Applied Biosystems)[57].

**Autoaggregation assay.** Briefly, the *V. cholerae* strains were grown overnight in LB media at 37 °C with agitation. The overnight cultures were centrifuged at $9000 \times g$, resuspended in DMEM media and normalized to an optical density at 600 nm (A600) of ~1.0. 50 mL of each bacterial suspension was placed in two separate tubes. One tube remained static and the other was vortexed before each OD measurement. The tubes were left static at room temperature. To measure the bacterial settling over time, at designated time points, 0.5 mL was removed from within 1 cm of the surface of each bacterial suspension and the A600 was measured. Experiments were independently performed three times.

**Statistics and reproducibility.** All data are expressed as the means ± standard deviation (SD). Differences between two groups were evaluated using a two-tailed unpaired Student's *t* test or Mann–Whitney *U* test according to the test requirements (as stated in the figure legends). Significance is indicated as *$P \leq 0.05$; **$P \leq 0.01$; ***$P \leq 0.001$ and n.s. means no significant difference. Statistical analysis of Mann–Whitney *U* test was conducted using the software MedCalc (v12.3.0.0). Statistical analysis of two-tailed unpaired Student's *t* test was conducted using the GraphPad Prism (v7.0.4). Biological triplicate experiments were conducted in all in vitro experiments. Mice colonization experiments were conducted twice with at least four mice in each group, and the combined data for the two experiments was used for statistical analysis.

**Reporting summary.** Further information on research design is available in the Nature Research Reporting Summary linked to this article.

## Data availability

The RNA-seq data have been deposited in the NCBI Sequence Read Archive database under accession codes SRR16991885 (HD-5), SRR16991886 (HD-5), SRR16991887 (HD-5), SRR16991888 (Control), SRR16991889 (Control) and SRR16991890 (Control). The ChIP-seq data have been deposited in the NCBI Sequence Read Archive database under accession codes SRR17012211 (ChIP-mock) and SRR17012212 (ChIP-CarR). Source data are available in Supplementary Data 3. All other data are available from the corresponding author on reasonable request.

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

## Acknowledgements
This work was funded by the National Key R&D Program of China 2021YFC2300300 (to K.Z.), the National Natural Science Foundation of China Grant 32100144 (to Y.L.), 81902030 (to T.X.), and 82172330 (to K.Z.); Science Foundation of Tianjin Grant 20JCQNJC01970 (to B.Y.); Fundamental Research Funds for the Central Universities, Nankai University Grant 980/63211149 (to B.Y.); Shenzhen Basic Research Key projects JCYJ20200109144220704 (to K.Z.); Shenzhen Basic Research projects JCYJ20190807144409307 (to K.Z.), and Shenzhen Science and Technology Program JCYJ20210324113608022 (to T.X.).

## Author contributions
B.Y., K.Z. designed the research; Y.L., T.X., Q.W., J.H., Y.Z., X.L. and R.L. performed the research; Y.L., Q.W., J.H. and Y.Z. contributed new reagents or analytic tools; T.X., J.H. and Y.Z. analyzed the data; Y.L., T.X., B.Y. and K.Z. wrote the paper.

## Competing interests
The authors declare no competing interests.
