## [Peer Review File · Communications Biology]

Reviewers' comments:

Reviewer #1 (Remarks to the Author):

(Please see also attachment)

The manuscript "Vibrio cholerae senses human enteric α -defensin 5 through a RstAB two-component system to promote bacterial pathogenicity" by Liu, Xu et al describes an interesting observation, namely that the RstAB two-component regulatory system of Vibrio cholerae detects the presence of numerous human-derived cationic peptides and that this detection is important both for induction of virulence-associated genes and for host colonization. I think this is interesting work, that is carefully performed and reported but I do have a few comments and concerns that I think might be helpful for the authors to address in a revised version of the manuscript.

The authors clearly show that RstAB is a regulator of Vibrio behaviour - consistent with previous work from Huang et al, 2018. They show that the system is activated by CAMPs and that regulation affects a large number of Vibrio genes. In this paper, they investigated a connection between the RstA gene and the TcpP virulence regulon, but I feel that in their discussion they might have missed an opportunity to highlight the large number of other targets of RstA that I think are very interesting as well. I don't think they need to expressly validate every single aspect of their HD5 and ChIP dataset, but I do think adding some of this to the discussion might enhance the impact and interest of the paper. I've added some other specific comments below.

Line by line comments below:

Line 144 (and figure 3) - I don't think the authors can conclude that a deficiency in small intestinal colonization is unrelated to growth. The data supports a model where there are differences in fitness, but it is unclear whether this is due to growth differences, adhesion differences or a combination of the two. In their CI assays, the authors only describe that they looked at small intestine associated bacteria - but they didn't separate tissue associated from luminal bacteria... this data might be valuable to this conclusion. Given that some of the HD5 and ChIP regulated genes include dicarboxylate transporters and other potentially important metabolic processes - this is an important consideration.

Line 164-5 - I understand why tcpP was chosen as a high priority target, but there are a large number of RstA binding sites throughout the genome that were barely discussed. I wonder about these other sites and if/whether they support a model of virulence where RstA is controlling more than TcpP-regulated virulence. To that end, I'd have liked some discussion/comparison of the HD5-regulated genes (as seen in Figure 1) and how many of these are also TcpP regulated and how many appear to be RstA unique. I'll return to this point later in discussing the CI experiments and the ChIP seq data.

Line 180-181 (and Figure 4h) - What is the limit of detection for the competition assay? I think that in order to make this conclusion, a competition assay between $\Delta rstA$ and $\Delta rstA/\Delta tcpP$ should be performed and there should be an attempt to compare both luminal contents and the tissue-associated contents (or at least to temper the conclusions drawn from the experiment performed in the discussion).

Line 202-205 - does the RstA mutant show altered aggregation in the absence of host cells? Are any of the target genes of RstA known adhesins?

All bar graphs - consider showing the underlying data points on the graphs... it would greatly help with interpretation

Fig 3D - is there a growth defect of $rstA^-$ in the presence of HD5?

Fig 3 F-G - While I like the precision of reporting P-values directly, these are hard to read (3f and 3g). Consider reporting p-values in the legend and using symbols in the figure. I wonder if these data might be represented in a single figure where HD5-/ + media results are shown on the same

graph? This clearly shows *rstA* dependent expression of *tcpP*, *tcpA*, *toxT* and *ctxA*, but it doesn't really show the effect of HD5 because normalization here is always to WT.
Fig 4B - these figures should show the gene name and the Vc number associated with the ChIP peak... it's very difficult to assess the results otherwise

Minor/typographical/visualization issues

Figure 1c - Up-regulated

Line 23 "where they produce..." suggests here that the bacteria produce CAMPs... rearrange the sentence so that it's clear these are produced by the host

Line 49 - the product of the *ctxAB* genes should be named

Line 49 - Rearrange sentence not to start with a lower case gene name

Line 69 - protein names are " human α -defensin 5 and human α -defensin 6" - no plural; should probably reference beta-defensins as well

Line 111 - Change to "Genes with unknown functions were not shown"

Line 127-228 - Can virulence be shown in vitro? By definition, it's an interaction between host and pathogen. Consider rephrasing to " Collectively, these results suggest 127 that *V. cholerae* may sense HD-5 to promote expression of virulence-associated genes"

Line 161-162 (and elsewhere) - consider adding the gene associated with each ChIP peak - it is very cumbersome to search through the supplemental table for the gene that these peaks correspond to... similarly - in the figure itself (Figure 4B) the authors should do the same.

Line 185 - Odd syntax

Line 188 - HD-6 is not directly antimicrobial - see Chairatana - 2017.

Line 192 - Odd syntax - consider "The AlmEFG proteins can modify lipid A by the addition of glycine and diglycine"

Line 196-197 - I don't understand this sentence... the data shows altered regulation of *rstAB* and *rstAB*-regulated genes in response to CAMPs... it says nothing about resistance to those peptides.

Line 239 - Consider revising to - " ...directly regulate the expression of *almEFG* to resist killing by polymyxin B"

Reviewer #2 (Remarks to the Author):

In the article *Vibrio cholerae* senses human enteric alpha defensin 5 through a RstAB two component system to promote bacterial pathogenicity, authors link defensin sensing TCS RstAB with increased pathogenicity via direct interaction of RstA and *tcpP* promoter.

In their work authors used cationic antimicrobial peptide human alpha-defensin 5 (HD-5) in RNA seq exp to seek differentially expressed genes in response to fixed concentration of compound in virulence inducing condition. Illumina RNA seq analysis identified 193 differentially expressed genes, and 10 genes in random were validated with qRT-PCR.
For the rest of the paper authors focused on characterizing link between HD-5 induced RstAB and

TcpP and *Vibrio Cholerae* pathogenicity trait, both in vivo and in vitro.

Conclusion and final model of this paper links direct influence of HD-5 on RstAB, which in turn directly bind to tcpP promoter, thus influencing *Vibrio cholerae* pathogenic traits.

Overall comments:

Authors has chosen to focus their work on regulation of TCP, mainly o tcpP promoter, but there are not investigating the mechanism of binding of RstA to this promoter, and potential interaction with other proteins regulating the expression of tcpP or discussion about properties of this promoter (divergent and even encoding small RNA). To proof direct binding of RstA to DNA one must characterize binding sequence and proof its specificity (e.g. mutation in DNA binding site which not allow RstA to bind to DNA).

Timing of induction of virulence gene expression is tightly regulated by quorum sensing. While expression of tcpP is maximal early on in vitro (3.5h in AKI condition), at this time point ctx is barely detectible. I am puzzled in which time point authors collected their samples for analysis (western, RNA seq or qRT PCR). It would be useful to get this information on all the experiments in material and methods, and make correlation between quorum sensing and expression of TCS.

TCP operon regulation is on autoregulatory loop, where TcpPH and ToxRS activate toxT expression. ToxT in turn activates tcpA and ctx genes coding for 2 major virulence factors. tcpP, tcpH, tcpA and toxT are part of TCP operon, while toxR is part of ancestral chromosome and has additional roles in live cycle of vibrio and it is not regulated by TcpPH. Fig 2 shows increased mRNA levels of tcpP, toxT, tcpA and toxR. Less than 2 fold increase of ctx production measured by western blot is contributed by BOTH arms of TCP regulation. This was not discussed by authors and contribution of ToxR is not accounted for /measured.

Perhaps the most perplexing questions to answer from this paper is a relationship between bacterial defense mechanisms against CAMPS and slight increase of virulence, one of the most energetically expensive step in the lifecycle of vibrio. From the RNA seq exp one can glance induction of genes involved in lipid A modification, LPS modification, induction of few components of ABC transport/efflux pumps, fatty acid regulation....suggesting that vibrio defense mechanism against HD-5 is being activates and HD-5 is being sequestered by OMV or actively pumped out of the cell. Their model (figure 6) is showing HD-5 direct interaction between RstB and CAMP at extracellular space, but RstB has a membrane fraction localized at inner membrane and C terminus localized in periplasm. RstA is predicted by PROSIT to be located at cytoplasm. Authors fail to show direct interaction between RstAB and HD-5 at extracellular space or in periplasm and fail to even speculate that remodeling of OM and/or changing murein structure could be sensed by RstAB TCS.

RstAB is also regulated by its activity (phosphorylation and autophosphorylation). Authors did not experiment/discuss this option and wild type version of the protein was used in EMSA experiments.

RNA seq analysis are missing statistical analysis, including P value and FDR (false discovery rate). It is hard to put any reasonable weight on results based solely on fold change.

Fold changes withing tcp operon are relatively small, highest one is app 4.6-fold upregulated in the presence of HD-5. No attention is paid to other genes (with reasonable fold change and – if run – better statistical values).

VC1318 and 1319 are transcriptionally induced in the presence of HD-5. RNA-seq does not measure activity of protein, but mRNA message. Authors fail to distinguish and discuss difference between those 2 events, or to look for the mechanism of transcriptional upregulation.

Article is surely missing a discussion about current RNA seq results and other RNA seq experiments with $\Delta rstA$ strain (as in reference 23). No attempt was made to discuss role/involvement of other HD-5 regulated genes (strong induction of VC1320 *OmpV*, located just next to VC1318 and VC1319).

Discussion is probably the weakest part of the paper. Model building figure 6 needs to link nuances

of transcriptional regulation of TCP, quorum sensing and small RNA regulation with another TCS activator responding to HD-5. Authors do not explain paradigm between potential killing of bacteria and enhancement of virulence gene expression or incorporation of existing data reporting that hBD-2, HD-5 and LL-37 peptides are normally present in the small intestine epithelium and amounts are decreasing at the acute stage of watery diarrhea.

Figure comments:

Figures are overcrowded. Some of them can be moved to supplement, some can be combined (CI exp in figure 3 and 4). qRT-PCR data are presented as a mean of n=3 biological replicates Did each biological replicate have its technical replicate? Western blot analysis are missing negative controls, so one can exclude cross reactive protein bands run at same place.

Fig1 legends do NOT correspond to graphs. Figure is overcrowded with information and some part can be moved to supplemented material.

Fig2 western blot using anti-CT antibody does not include Δ ctx strain as a negative control, it is hard to determine specificity of this antibody. Also, there is no time mention in which samples were collected for this analysis.

Fig2 qRT-PCR analysis shows increased levels of all 4 transcripts (tcpP, tcpA, toxT, toxR) including toxR. toxR is expressed from an operon that is not associated with either the VPI or CTX elements and not regulated by ToxT or TcpPH.

Fig2 growth of *Vibrio Cholerae* : CFU would be good indication of viability in supplement.

Fig3 is overcrowded and can be split into more figures or some of info can be put into supplemented material. Western blot does not contain negative control (e.g. Δ ctx strain). Using very light band of RNAP as a loading control does not serve its purpose and different, well expressed protein can serve a better, more reliable control.

Fig4 overcrowded and some of the data can be put into supplement. In vivo competition graph should include Δ rstA as well, so reader would not need to rely on Fig 3h. It is hard to establish, if Δ rstA and Δ tcpP and Δ rstA Δ tcpP mutants are having save colonization defect.

At Fig 4; d-g, EMSA experiments: Wild type RstA with His6 on C terminal protein was incubated with 4 different pieces of labeled DNA to determine direct binding of wt-6His protein to different promoters in concentration from 0-800nM. Since RstA needs to be phosphorylated in order to bind DNA, not too much conclusion can be made from this experiment. In order to properly execute this experiment, phosphorylated RstA or wt RstA with a point mutation mimicking phosphorylated form needs to be used. To determine specificity, competition with "cold" DNA needs to be used as a control. It is VERY unusual to see to use 800nM of transcriptional regulatory protein in EMSA and not to see complete gel shift.

Legend comments:

Western blotting – line 308: which gel was used? 4-20% or 12% SDS PAGE?

In conclusion – introduction of current model of regulation of TCP via direct binding of RstA to tcpP promoter in HD-5 dependent manner needs to close few technical holes (stat analysis of RNA seq exp, inclusion of negative controls in western blots, inclusion of "cold" (competitive) DNA in EMSA experiments, using activated form of RstA in EMSA, establishing protein binding sequence.

Article would benefit from different structure. If authors would chose to still focus on TCP regulation, more attention should be paid to structure of tcpP promoter, position of binding of RstA to reference of transcriptional start of tcpP, tcpI or small RNA tarA and overlapping binding site with different transcriptional activators/repressors.

I would strongly recommend to strengthen paper by including more experiments, reanalyzing data, restructuring the flow and rewriting discussion. Refining the model with proven data could shed a new light into colonization and virulence of *vibrio cholerae* and be beneficial for the field.

Reviewer #3 (Remarks to the Author):

Liu et al investigate the response of the cholera pathogen to the human enteric alpha-defensin 5 and find that this peptide promotes *V. cholerae* virulence gene expression by triggering the pathogen's RstAB two component system. They provide evidence that RstA directly activates virulence gene expression by binding to the promoter of the virulence regulator TcpP. In general, the data is convincing, and it provides new knowledge of a molecular pathway by which host factors may trigger the pathogen's virulence program.

The observations that RstA promotes virulence gene expression is new but the introduction of this paper does not place this data within the context of the literature. Reference 37, mentioned here only in the discussion, discovered Vc 1319-20 at the same time as ref 23 and named the RstAB two component system VprAB and showed that its expression was increased by the cationic peptide polymyxin (a similar agent as HD-5). Moreover, both ref 23 and 37 showed that an RstA mutant has a colonization defect. The current paper raises the possibility the colonization defect is due to diminished virulence gene expression, rather than defective lipid A modification, but this idea is not discussed. This may be true because Paneth cells are not mature in 5-day old mice and there is little production of antimicrobial peptides at this point. Thus, it is not possible to use this model to understand *V. cholerae* response to CAMPs in vivo. In any case, their conclusion in the abstract that 'this study established the colonization site recognition mechanism of *V. cholerae*' is an extreme over-statement and should be toned considerably or deleted.

General comments

1. RstAB is the name of genes in the cholera toxin prophage in *V. cholerae* and it would be preferable if the name CarSR was used instead.
2. Figure 1:
 - A. Showing the RNA-seq data on circos plots is not informative; generally MA plots are used to compare transcriptional profiles.
 - B. Are the COG categories listed significantly enriched or depleted?
3. The Fig1 legend does not correspond to the panels in the figure; e.g. legend 1c describes panel a.
3. Fig 2 and 3, the use of the abbreviation of CT for control (because CT often means cholera toxin) is confusing spell out 'control' or use 0 HD-5.
4. Fig 3 and 4, For the CI assays note should be made in the legends that competitions are vs a lacZ- WT strain
5. Figure 4
 - A. Its hard to see the shifted band for almE.
 - B. Note that binding of RstA to the tcpH promoter does not prove that RstA activates tcpH transcription.
6. Line 205-206 CAco-2 adherence is not equivalent to 'colonization'.
7. Line 228-229 the claim that they show that *V. cholerae* senses HD-5 in the crypt to confer a colonization advantage in the small intestine' is a unwarranted. There is no data shown here to show that d5 mice express defensins.
6. Ref 27 shows that defective colonization in a CarR (RstA) mutant is strain dependent; the authors should mention this fact in the discussion.
7. In general, the English is understandable but the manuscript still needs some polishing.

Minor points

1. The methods should state the origin of the *V. cholerae* strain e.g. year of isolation and biotype.
2. The methods should state the sources of the HD-5, HD-6, HBD2 and LL-37.

Reviewers' comments:

Reviewer #1 (Remarks to the Author):

The manuscript "Vibrio cholerae senses human enteric α -defensin 5 through a RstAB two-component system to promote bacterial pathogenicity" by Liu, Xu et al describes an interesting observation, namely that the RstAB two-component regulatory system of Vibrio cholerae detects the presence of numerous human-derived cationic peptides and that this detection is important both for induction of virulence-associated genes and for host colonization. I think this is interesting work, that is carefully performed and reported but I do have a few comments and concerns that I think might be helpful for the authors to address in a revised version of the manuscript.

The authors clearly show that RstAB is a regulator of Vibrio behaviour - consistent with previous work from Huang et al, 2018. They show that the system is activated by CAMPs and that regulation affects a large number of Vibrio genes. In this paper, they investigated a connection between the RstA gene and the TcpP virulence regulon, but I feel that in their discussion they might have missed an opportunity to highlight the large number of other targets of RstA that I think are very interesting as well. I don't think they need to expressly validate every single aspect of their HD5 and ChIP dataset, but I do think adding some of this to the discussion might enhance the impact and interest of the paper. I've added some other specific comments below.

Response:

Thank you for your great patience in listing the problems that exist with the manuscript. We have carefully revised the manuscript based on your comments and suggestions. Especially, several parts in the Discussion sections have been rewritten.

Detailed point-by-point responses are provided below.

Line by line comments below:

1. Line 144 (and figure 3) - I don't think the authors can conclude that a deficiency in small intestinal colonization is unrelated to growth. The data supports a model where

there are differences in fitness, but it is unclear whether this is due to growth differences, adhesion differences or a combination of the two. In their CI assays, the authors only describe that they looked at small intestine associated bacteria - but they didn't separate tissue associated from luminal bacteria... this data might be valuable to this conclusion. Given that some of the HD5 and ChIP regulated genes include dicarboxylate transporters and other potentially important metabolic processes - this is an important consideration.

Response:

Thanks for the comment. Competitive infection assays have now been performed to measure bacterial CFU from infant mice infected with WT, $\Delta carR$ ($\Delta rstA$ of original MS) and $\Delta carR+$ ($\Delta rstA+$ of original MS) to assess bacterial abundance from the luminal content and epithelium in the small intestine at 1 d post-infection. The results showed that the CI values of the $\Delta carR$ and $\Delta carR+$ versus WT strain in the luminal content were 1.09 and 1.22, respectively, while the CI values in the epithelium were 0.38 and 1.19. These results indicated that CarR (RstA) promotes *V. cholerae* colonization in the mouse small intestine by increasing adhesion to the intestinal epithelium, but not by increasing bacterial growth in the intestinal lumen. The relevant information is now described in the Results of the revised manuscript (pg. 6-7, lines 149-154; Fig. 4a).

2. Line 164-5 - I understand why *tcpP* was chosen as a high priority target, but there are a large number of RstA binding sites throughout the genome that were barely discussed. I wonder about these other sites and if/whether they support a model of virulence where RstA is controlling more than TcpP-regulated virulence. To that end, I'd have liked some discussion/comparison of the HD5-regulated genes (as seen in Figure 1) and how many of these are also TcpP regulated and how many appear to be RstA unique. I'll return to this point later in discussing the CI experiments and the ChIP seq data.

Response:

Thank you for your comments. ChIP-Seq results showed that in addition to binding to *tcpP* promoter, CarR also binds to several other promoters of genes. These genes include genes encoding putative hemolysin (*vc0489*, *vca0218*, *vca0219*), encoding

regulators (*vc1021*; *vc1617*; *vc1741*; *vca1078*), encoding adhesin factors (*vc1318*; *vc0633*), encoding transporters (*vc0069*; *vc1325*; *vc1669*; *vca0205*; *vca0227*), encoding amino acid ligase (*vc1579*) and encoding tryptophanase (*vca0161*, *tnaA*). The presence of a large number of binding sites indicates that CarR plays multiple regulatory roles. These contents have now been added to the Discussion section of the revised MS (lines 323-329).

Combined analysis of RNA-seq and ChIP-seq data revealed that some other up-regulated genes induced by HD-5 were also identified in ChIP-seq results. These genes include *vc1318(ompV)* encoding putative adhesin, *vc1579(almE)* encoding lipid A modification system glycine--protein ligase, *vc1727(glgC)* encoding glucose-1-phosphate adenylyltransferase and *vc1741* encoding a regulator. These results indicated that *V. cholerae* CarR directly regulates the expression of genes involved in antimicrobial resistance (*almE*), adhesin (*ompV*), metabolism pathway (*glgC*) and regulation pathway (*vc1741*) in response to HD-5. Furthermore, considering all these genes are not under the control of TcpP, thus CarR regulates the expression of these genes uniquely and independent of TcpP. These contents have now been added to the Discussion section of the revised MS (lines 329-338 of the revised MS).

3. Line 180-181 (and Figure 4h) - What is the limit of detection for the competition assay? I think that in order to make this conclusion, a competition assay between Δ rstA and Δ rstA/ Δ tcpP should be performed and there should be an attempt to compare both luminal contents and the tissue-associated contents (or at least to temper the conclusions drawn from the experiment performed in the discussion).

Response:

Thank you for your comments. We have now constructed a double mutant Δ carR *lacZ*- as parent strain and performed additional experiments to show the CI value of Δ carR Δ tcpP versus Δ carR *lacZ*- was 1.09 and 0.285 in luminal contents and the tissue-associated contents of mice, which was similar to the CI value (1.22 and 0.327) of Δ tcpP versus Δ carR *lacZ*- in the luminal contents and the tissue-associated contents (Fig. 5h of the revised MS). These results suggest that the influence of TcpP on the colonization of *V. cholerae* is mediated by CarR. These contents have now been added to the Results section of the revised MS (lines 209-216 of the revised MS).

4. Line 202-205 - does the RstA mutant show altered aggregation in the absence of host cells? Are any of the target genes of RstA known adhesins?

Response:

Thank you for your comments. Autoaggregation assays have now been performed to determine the effect of CarR on the autoaggregation capacity of *V. cholerae* in DMEM media according to a previously described method (*Sci Rep.* 2017 Aug 1;7(1):7011.). The result showed that the A600 values of WT and $\Delta carR$ decreased substantially after 1h incubation at 37°C in DMEM media (Supplementary Fig. 4a in revised MS). This data confirmed that *carR* regulates adherence capacity via modulating virulence gene expression, rather than by regulating autoaggregation capacity. These contents have now been added to the Results and Methods section of the revised MS (lines 243-250, 506-515 of the revised MS).

ChIP-seq results showed that two CarR binding sites were located in the promoter regions of *ompU* (*vc0633*) and *ompV* (*vc1318*), encoding adhesin factors of *V. cholerae*. This suggests that CarR may increase the adherence capacity of *V. cholerae* through directly activating the expression of *ompU* and/or *ompV*. However, RNA-seq results showed that HD-5 had no effect on the expression of *ompU*, indicating that *ompU* were not involved in HD-5-mediated virulence regulatory pathway. Moreover, *ompV* was significantly up-regulated in response to HD-5. This indicated that CarR can directly promote *ompV* expression, which may enhance *V. cholerae* adherence capacity in response to HD-5. These contents have now been added to the Discussion section of the revised MS (lines 329-338 of the revised MS).

5. All bar graphs - consider showing the underlying data points on the graphs... it would greatly help with interpretation

Response:

Thank you for your suggestion. The underlying data points on the graphs have been added as suggested.

6. Fig 3D - is there a growth defect of *rstA*- in the presence of HD5?

Response:

Thank you for your comment. Bacterial growth curves in AKI medium supplemented with 50 µg/mL HD-5 showed that $\Delta carR$ exhibited a lower growth rate and reached a lower final cell counts than WT (Fig. 4e in revised MS). This result suggested that CarR enhances resistance to HD-5, thereby maintaining the normal growth of *V. cholerae*. Therefore, CarR regulates both the virulence and HD-5 resistance in *V. cholerae*. These contents have now been added to the Results section of the revised MS (lines 161-166 of the revised MS).

7. Fig 3 F-G - While I like the precision of reporting P-values directly, these are hard to read (3f and 3g). Consider reporting p-values in the legend and using symbols in the figure. I wonder if these data might be represented in a single figure where HD5-/+ media results are shown on the same graph? This clearly shows *rstA* dependent expression of *tcpP*, *tcpA*, *toxT* and *ctxA*, but it doesn't really show the effect of HD5 because normalization here is always to WT.

Response:

Thank you for your suggestions. The p values have now been changed to asterisk (*) in all figures. The calculation method of p-value and the meaning of each symbol has been added in the legends of each figure. Fig. 3f and 4a in original MS have been represented in a single figure (Fig. 4b in revised MS) as suggested.

8. Fig 4B - these figures should show the gene name and the Vc number associated with the ChIP peak... it's very difficult to assess the results otherwise

Response:

Thank you for your comments. The peak names have now been changed to gene names in Fig. 5a of the revised MS (lines 172-173 of the revised MS).

Minor/typographical/visualization issues

1. Figure 1c - Up-regulated

Response:

Revised as suggested.

2. Line 23 "where they produce..." suggests here that the bacteria produce CAMPs... rearrange the sentence so that it's clear these are produced by the host

Response:

The sentence "When *V. cholerae* is ingested, the bacteria colonize the epithelium of the small intestine, where they produce large amounts of cationic antimicrobial peptides (CAMPs)." in the original manuscript (pg. 2, lines 23-25) has been modified as "When *V. cholerae* is ingested, the bacteria colonize the epithelium of the small intestine and stimulate the Paneth cells to produce large amounts of cationic antimicrobial peptides (CAMPs)." in the revised manuscript (pg. e2, lines 22-24).

3. Line 49 - the product of the *ctxAB* genes should be named

Response:

Done as suggested (pg. e3, line 50).

4. Line 49 - Rearrange sentence not to start with a lower case gene name

Response:

The sentence "*toxT* is regulated by two integral membrane regulatory proteins, ToxR and TcpP" in the original manuscript (pg. 3, lines 49) has been modified as "The transcription of *toxT* is regulated by two integral membrane regulatory proteins, ToxR and TcpP." in the revised manuscript (pg. 3, lines 50-51).

5. Line 69 - protein names are " human α -defensin 5 and human α -defensin 6" - no plural; should probably reference beta-defensins as well

Response:

The sentence " α -defensins 5, α -defensins 6" in the original manuscript (pg. 3, lines 69) has been modified as " α -defensins, β -defensins" in the revised manuscript (pg. 4, lines 70).

6. Line 111 - Change to "Genes with unknown functions were not shown"

Response:

The sentence "Unknown functions were not shown." in the original manuscript (pg. 5, lines 111) has been modified as "Genes with unknown functions were not shown." in the revised manuscript (pg. 5, line 113).

7. Line 127-128 - Can virulence be shown *in vitro*? By definition, it's an interaction between host and pathogen. Consider rephrasing to "Collectively, these results suggest that *V. cholerae* may sense HD-5 to promote expression of virulence-associated genes"

Response:

The sentence "Collectively, these results suggest that *V. cholerae* may sense HD-5 to promote virulence *in vitro*." in the original manuscript (pg. 5, lines 127-128) has been modified as "Collectively, these results suggest that *V. cholerae* may sense HD-5 to promote expression of virulence-associated genes." in the revised manuscript (pg. 6, lines 131-132).

8. Line 161-162 (and elsewhere) - consider adding the gene associated with each ChIP peak - it is very cumbersome to search through the supplemental table for the gene that these peaks correspond to... similarly - in the figure itself (Figure 4B) the authors should do the same.

Response:

Done. Each ChIP peak names have now been changed to the corresponding gene names as suggested. (Lines 172-173 and Fig. 5a of the revised manuscript).

9. Line 185 - Odd syntax

Response:

The sentence "Antimicrobial peptides exacerbate colonization and virulence gene expression is widespread in *V. cholerae*" in the original manuscript (pg. 7, lines 185-

186) has been modified as “CAMPs promote *V. cholerae* adhesion and virulence gene expression through similar molecular mechanisms.” in the revised manuscript (pg. 8, lines 218-219).

10. Line 188 - HD-6 is not directly antimicrobial - see Chairatana -2017.

Response:

The sentence “HD6 lacks the broad-spectrum antimicrobial activity observed for other human α -defensins.” has now been added in revised MS. (Line 223-225 in revised MS)

11. Line 192 - Odd syntax - consider "The AlmEFG proteins can modify lipid A by the addition of glycine and diglycine"

Response:

The sentence “The AlmEFG proteins can modify the glycine and diglycine of lipid A” in the original manuscript (pg. 8, lines 192) has been modified as “The AlmEFG proteins can modify lipid A by the addition of glycine and diglycine” in the revised manuscript (pg. 8, lines 226-227).

12. Line 196-197 - I don't understand this sentence... the data shows altered regulation of *rstAB* and *rstAB*-regulated genes in response to CAMPs... it says nothing about resistance to those peptides.

Response:

We are sorry for the confusion of the relevant statement in the original MS. The results showed that all these CAMPs can up-regulate the expression of *carSR* and *almE*, which was previously shown as a CarSR regulated gene in resistance to Polymyxin B as a positive control. The statement ‘These results indicate that CarSR positively responds to CAMPs.’ has been changed to “These results indicate that different CAMPs can both active expression of *carSR* and downstream genes in vitro.” . (Line 233-235 in revised MS)

13. Line 239 - Consider revising to - "...directly regulate the expression of *almEFG* to resist killing by polymyxin B"

Response:

The sentence "In *V. cholerae*, RstAB was identified to sense polymyxin B and directly regulate the expression of *almEFG* to resistance to polymyxin B." in the original manuscript (pg. 9, lines 238-240) has been modified as "In *V. cholerae*, CarSR was identified to sense polymyxin B and directly regulate the expression of *almEFG* to resist killing by polymyxin B." in the revised manuscript (pg. 11, lines 278-280).

Reviewer #2 (Remarks to the Author):

In the article *Vibrio cholerae* senses human enteric alpha defensin 5 through a RstAB two component system to promote bacterial pathogenicity, authors link defensin sensing TCS RstAB with increased pathogenicity via direct interaction of RstA and *tcpP* promoter.

In their work authors used cationic antimicrobial peptide human alpha-defensin 5 (HD-5) in RNA seq exp to seek differentially expressed genes in response to fixed concentration of compound in virulence inducing condition. Illumina RNA seq analysis identified 193 differentially expressed genes, and 10 genes in random were validated with qRT-PCR.

For the rest of the paper authors focused on characterizing link between HD-5 induced RstAB and *TcpP* and *Vibrio Cholerae* pathogenicity trait, both in vivo and in vitro.

Conclusion and final model of this paper links direct influence of HD-5 on RstAB, which in turn directly bind to *tcpP* promoter, thus influencing *Vibrio cholerae* pathogenic traits.

Response:

Thank you for your patient and thoughtful reading as well as the constructive comments and advices about our manuscript. We have revised the manuscript based on your comments and suggestions. Especially, several parts in the Discussion

sections have been rewritten.

Detailed point-by-point responses are provided below.

Overall comments:

1. Authors has chosen to focus their work on regulation of TCP, mainly on *tcpP* promoter, but there are not investigating the mechanism of binding of RstA to this promoter, and potential interaction with other proteins regulating the expression of *tcpP* or discussion about properties of this promoter (divergent and even encoding small RNA). To proof direct binding of RstA to DNA one must characterize binding sequence and proof its specificity (e.g. mutation in DNA binding site which not allow RstA to bind to DNA).

Response:

Thank you for your comments. We have performed electrophoretic mobility shift assays (EMSA) and competition assays. The results showed that at increasing concentrations of phosphorylated CarR (RstA in original MS) protein, slow migrating bands were observed for the FAM-labeled promoters of *tcpP* and *almE* (positive control). Moreover, addition of unlabeled promoters effectively competed for CarR binding to the labeled promoters, and the retarded band disappeared in the presence of 100-fold excess unlabeled promoter DNA (Fig. 5d and Supplementary Fig. 2a). These results indicate that phosphorylated CarR binds specifically to the promoter regions of *tcpP* and *almE* *in vitro*. In addition, the binding capacity of non-phosphorylated CarR to the promoter regions of *tcpP* and *almE* was significantly reduced compared with that of phosphorylated CarR (Fig. 5e and Supplementary Fig. 2b). Meanwhile, phosphorylated CarR did not bind to the negative control (the promoter regions of *rpoS* and *toxT*) under the same experimental conditions (Supplementary Fig. 2c, d). The relevant information is now described in the Results of the revised manuscript (pg.7-8, lines 184-196).

To identify the CarR binding site in the promoter region of *tcpP*, we performed a dye-based DNase I foot-printing assay and found that CarR binds to a specific sequence containing a 45-bp motif, which is located -198 to -152 from the translational start site (Fig. 5c in revised MS). Further sequence analysis showed that a potential RstA box

(5-TAATGAGAATTACT-3) from -186 to -172bp was found. To further determine whether the 45-bp motif and the potential RstA box are important for binding to CarR, we performed EMSA and competition assays again using a P_{tcpP-1} DNA fragment (without the 45-bp motif) and a P_{tcpP-2} DNA fragment (without the potential RstA box) under the same conditions (Fig. 5f, g in revised MS). Neither the P_{tcpP-1} nor P_{tcpP-2} DNA fragment was able to bind to CarR, confirming that the potential RstA box is crucial to the binding ability of CarR to P_{tcpP}. The relevant information is now described in Results, Discussion and Methods section of the revised MS (lines 197-207, 312-317, 494-505 of the revised MS).

Previous results showed that AphA, AphB, HapR and CRP can directly bind to *tcpP* promoter (*Front Microbiol.* 2020 Apr 17; 11:709.; *Mol Microbiol.* 2001 Jul;41(2):393-407.). In addition, a small RNA (~91 nt), which was named TarA, is located in *tcpP* promoter (*Mol Microbiol.* 2010 Dec;78(5):1171-81.). However, the CarR binding site in *tcpP* promoter does not overlap with the binding sequence of AphA, AphB, HapR, CRP and coding sequence of *tarA* (Supplementary Fig. 3a). The relevant information is now described in Discussion of the revised MS (lines 317-323).

2. Timing of induction of virulence gene expression is tightly regulated by quorum sensing. While expression of *tcpP* is maximal early on in vitro (3.5h in AKI condition), at this time point *ctx* is barely detectible. I am puzzled in which time point authors collected their samples for analysis (western, RNA seq or qRT PCR). It would be useful to get this information on all the experiments in material and methods, and make correlation between quorum sensing and expression of TCS.

Response:

Thank you for your comments. For western blotting, RNA-seq and qRT-PCR experiments, the *V. cholerae* was grown in LB medium overnight and diluted 1:100 in fresh AKI medium. After anaerobically incubation at 37 °C to reach an OD₆₀₀ of 0.2, HD-5 was added at a concentration of 50 µg/mL. The bacterial culture was further incubated to an OD₆₀₀ of 1.0 with shaking, and the culture was then collected for subsequent western, RNA seq and qRT PCR experiments. This information has now been added to the Methods section of the revised MS (lines 355-358, 374-376,

413-415 of the revised MS).

3. TCP operon regulation is on autoregulatory loop, where TcpPH and ToxRS activate *toxT* expression. ToxT in turn activates *tcpA* and *ctx* genes coding for 2 major virulence factors. *tcpP*, *tcpH*, *tcpA* and *toxT* are part of TCP operon, while *toxR* is part of ancestral chromosome and has additional roles in live cycle of vibrio and it is not regulated by TcpPH. Fig 2 shows increased mRNA levels of *tcpP*, *toxT*, *tcpA* and *toxR*. Less than 2 fold increase of *ctx* production measured by western blot is contributed by BOTH arms of TCP regulation. This was not discussed by authors and contribution of ToxR is not accounted for /measured.

Response:

Thank you for your comments. Indeed, we made a mistake in Fig. 2b in original MS. When generating the Fig. 2b in original MS, we forgot to change the gene names in primary templates. After careful examination, we found that the *toxR* and *tcpA* in Fig. 2b in original MS should be *tcpA* and *ctxA*. we have fixed the figure accordingly in Fig. 2b in revised MS. Furthermore, we have performed additional qRT-PCR experiment to analysis the expression of *toxR*. The results showed that expression of *toxR* exhibited no significant change in the presence of HD-5. RNA-seq results also showed that *toxR* was up-regulated only 1.2-fold in the presence of HD-5. This indicated that HD-5 induced TCP operon up-regulation is mediated by *tcpP* rather than *toxR*. This information has now been added to the Results and Discussion sections of the revised MS (lines 123-124 of the revised MS).

4. Perhaps the most perplexing questions to answer from this paper is a relationship between bacterial defense mechanisms against CAMPS and slight increase of virulence, one of the most energetically expensive step in the lifecycle of vibrio. From the RNA seq exp one can glance induction of genes involved in lipid A modification, LPS modification, induction of few components of ABC transport/efflux pumps, fatty acid regulation....suggesting that vibrio defense mechanism against HD-5 is being activates and HD-5 is being sequestered by OMV or actively pumped out of the cell. Their model (figure 6) is showing HD-5 direct interaction between RstB and CAMP at extracellular space, but RstB has a memnilaizbrane fraction localized at inner membrane and C terminus localized in periplasm. RstA is predicted by PROSIT to be

located at cytoplasm. Authors fail to show direct interaction between RstAB and HD-5 at extracellular space or in periplasm and fail to even speculate that remodeling of OM and/or changing murein structure could be sensed by RstAB TCS.

Response:

Thank you for your comments. We have now provided a model for to link nuances of transcriptional regulation of TCP, quorum sensing, small RNA regulation and antimicrobial resistance regulatory pathway according to your suggestions. The model has now shown in Figure 7 in revised MS. When *V. cholerae* enters the small intestine at initial stage of colonization, quorum sensing (QS) is not induced. Four small RNAs called Qrr1-4 activate translation of AphA and inhibit production of HapR. Meanwhile, cAMPs in small intestine interact with the lipopolysaccharide of *V. cholerae* and then gaining access to the periplasmic space. This process has mentioned in several research (*FEMS Microbiol Lett.* 2012 May;330(2):81-9.; *Front Microbiol.* 2014 Nov 26;5:643.; *J Microbiol.* 2020 Dec;58(12):979-987. *Infect Immun.* 2015 Mar; 83(3): 1199–1209.). Then, CarS was activated by periplasmic cAMPs and activates the phosphorylated state of CarR in the cytoplasm. Consequently, expression of *tcpP* and *almE* are upregulated, thereby promoting bacterial colonization and antimicrobial resistance.

During later stages of infection, as the population density increases, the increased cellular levels of cAMP induce the efflux water into the lumen of the small intestine that causes watery diarrhea. At this stage, CarS was inactivated because of decreased cAMPs concentration and HapR was induced at high cell density. During these process, *V. cholerae* down-regulates the expression of virulence factors and detach to disseminate throughout the small intestine. This relative information has now been added to the Discussion sections of the revised MS (lines 253-268 and Fig. 7 in revised MS).

5. RstAB is also regulated by its activity (phosphorylation and autophosphorylation). Authors did not experiment/discuss this option and wild type version of the protein was used in EMSA experiments.

Response:

Thank you for your comments. We have re-performed EMSA and competition assays using phosphorylated CarR and FAM-labeled DNA probes. CarR is phosphorylated by acetyl phosphate in vitro. The results showed that at increasing concentrations of phosphorylated CarR (RstA in original MS) protein, slow migrating bands were observed for the FAM-labeled promoters of *tcpP* and *almE* (positive control). Moreover, addition of unlabeled promoters effectively competed for CarR binding to the labeled promoters, and the retarded band disappeared in the presence of 100-fold excess unlabeled promoter DNA (Fig. 5d and Supplementary Fig. 2a). These results indicate that phosphorylated CarR binds specifically to the promoter regions of *tcpP* and *almE* in vitro. In addition, the binding capacity of non-phosphorylated CarR to the promoter regions of *tcpP* and *almE* was significantly reduced compared with that of phosphorylated CarR (Fig. 5e and Supplementary Fig. 2b). Meanwhile, phosphorylated CarR did not bind to the negative control (the promoter region of *toxT* and *rpoS*) under the same experimental conditions (Supplementary Fig. 2c, d). The relevant information is now described in the Results of the revised manuscript (pg.7-8, lines 184-196).

6. RNA seq analysis are missing statistical analysis, including P value and FDR (false discovery rate). It is hard to put any reasonable weight on results based solely on fold change.

Response:

Thank you for your comments. We have now added the P value and FDR in revised Supplementary Data 1. Differential gene expression have been re-analyzed, and expression changes >2-fold and P values ≤ 0.05 were considered statistically significant. The results showed that 123 genes were differentially expressed, with 41 downregulated and 82 upregulated. The relevant information is now described in the Results of the revised manuscript (pg. 4, lines 95-98; Fig. 1a, Supplementary Data 1).

7. Fold changes withing *tcp* operon are relatively small, highest one is app 4.6-fold upregulated in the presence of HD-5. No attention is paid to other genes (with reasonable fold change and – if run – better statistical values).

Response:

According to your suggestions, we further analyzed the RNA-seq results. The results showed that the expression of several genes was significantly induced. The products of these genes are mainly involved in lipid A modification (*vc1579*), lipopolysaccharide biosynthesis (*vc0235*), oligopeptide ABC transporter (*vc1092*, *vc1093*), amino acid ABC transporter (*vc1359*; *vc1863*; *vc1864*), long-chain fatty acid transport(*vca0862*), putative adhesion(*vc1318*) and transcription regulation (*vc1741*, *vca0642* and *vca0704*). These results suggest that *V. cholerae* greatly promotes the stress defense capacity of *V. cholerae* against HD5 by sensing HD-5 and regulating the expression of multiple genes simultaneously. For instance, *V. cholerae* may promote the expression of genes encoding ABC transporters to pump HD-5 out of the bacterial cells or modify LPS-lipid A to confer HD-5 resistance. This information has now been added to the Discussion sections of the revised MS (lines 289-298 of the revised MS).

8. VC1318 and 1319 are transcriptionally induced in the presence of HD-5. RNA-seq does not measure activity of protein, but mRNA message. Authors fail to distinguish and discuss difference between those 2 events, or to look for the mechanism of transcriptional upregulation.

Response:

Thank you for your comments. According to your suggestion, we have now performed additional western blotting assays to analyze the production of CarR in AKI medium supplemented with 0 or 50 µg/mL HD-5. Western blotting assays showed that CarR protein level exhibited a significant increase in AKI medium supplemented with 50 µg/mL HD-5(Fig. 3d in revised MS). Moreover, we have now performed additional analyses to determine the phosphorylation level of CarR in AKI medium supplemented with 50 µg/mL HD-5. The result showed that the phosphorylation level of CarR in AKI medium supplemented with 50 µg/mL HD-5 was significantly increased compared to that of 0 µg/mL HD-5(Fig. 3e in revised MS). These results indicated that HD-5 promotes *carR* expression both at transcriptional level and translational level and enhances CarR activity by increasing phosphorylation levels. These results have now been added to the revised MS (lines 140-145 of the revised MS).

9. Article is surely missing a discussion about current RNA seq results and other RNA seq experiments with $\Delta rstA$ strain (as in reference 23). No attempt was made to discuss role/involvement of other HD-5 regulated genes (strong induction of VC1320 *OmpV*, located just next to VC1318 and VC1319).

Response:

Thank you for your comments. According to your suggestion, we have now made a comparison of previous $\Delta carR$ RNA-seq results with our HD-5 RNA-seq results. The results showed that *vc1579(almE)* and *vc1318(ompV)* were significantly down-regulated in $\Delta carR$ RNA-seq results but significantly up-regulated in HD-5 RNA-seq results. *almE* has been proved positively regulated by *carR* to promote polymyxin B resistance. Meanwhile, *ompV* can help in adhesion and invasion of bacteria to intestinal epithelial cells and play a vital role in the pathogenesis of *S. Typhimurium*. The mechanism of *ompV* in *V. cholerae* need further investigate. This information has now been added to the Discussion sections of the revised MS (lines 298-306 of the revised MS).

10. Discussion is probably the weakest part of the paper. Model building figure 6 needs to link nuances of transcriptional regulation of TCP, quorum sensing and small RNA regulation with another TCS activator responding to HD-5. Authors do not explain paradigm between potential killing of bacteria and enhancement of virulence gene expression or incorporation of existing data reporting that hBD-2, HD-5 and LL-37 peptides are normally present in the small intestine epithelium and amounts are decreasing at the acute stage of watery diarrhea.

Response:

Thank you for your comments. We have now provided a model for to link nuances of transcriptional regulation of TCP, quorum sensing and small RNA regulation with virulence regulatory pathway, as shown in Figure 7. Briefly, when *V. cholerae* enters the small intestine at initial stage of colonization, quorum sensing (QS) is not induced. Four small RNAs called Qrr1-4 activate translation of AphA and inhibit production of HapR. Meanwhile, CAMPs in small intestine interact with the lipopolysaccharide of *V. cholerae* and then gaining access to the periplasmic space. This process has

mentioned in several research (*FEMS Microbiol Lett.* 2012 May;330(2):81-9.; *Front Microbiol.* 2014 Nov 26; 5:643.; *J Microbiol.* 2020 Dec;58(12):979-987. *Infect Immun.* 2015 Mar; 83(3): 1199–1209.). CarS was activated by periplasmic cAMPs and activates the phosphorylated state of CarR. Consequently, expression of *tcpP* and *almE* are upregulated, thereby promoting bacterial colonization and antimicrobial resistance.

During later stages of infection, as the population density increases, the increased cellular levels of cAMP induces the efflux water into the lumen of the small intestine that causes watery diarrhea. At this stage, CarS was inactivated because of decreased cAMPs concentration and HapR was induced at high cell density. During these process, *V. cholerae* down-regulates the expression of virulence factors and detach to disseminate throughout the small intestine. This information has now been added to the Discussion sections of the revised MS (lines 253-268 of the revised MS).

1. Figures are overcrowded. Some of them can be moved to supplement, some can be combined (CI exp in figure 3 and 4). qRT -PCR data are presented as a mean of n=3 biological replicates Did each biological replicate have its technical replicate? Western blot analysis are missing negative controls, so one can exclude cross reactive protein bands run at same place.

Response:

Thank you for your comments. Fig. 1c, Fig. 4f, g, original sequence peaks according to the ChIP-seq analyses in Fig. 4d-g of the original manuscript have been removed into Supplementary material (Supplementary Fig. 1a, Supplementary Fig. 2a-e of the revised manuscript).

The qRT -PCR results were calculated from 3 independent experiments with 3 technical replicates per experiment. This information has now been added to the Methods sections of the revised MS (lines 399-400 of the revised MS).

The Δ *ctxAB* strain has been constructed and used as negative control for Western blotting. The results showed no obvious bands for the cholera toxin were evident in Δ *ctxAB* strain. (Fig. 4d in revised MS).

2. Fig1 legends do NOT correspond to graphs. Figure is overcrowded with

information and some part can be moved to supplemented material.

Response:

Thank you for your comments. (1). Fig. 1 legend (original MS) has been corrected in revised MS. (2). Fig. 1a has now changed to MA plots in revised MS. (3). Fig. 1c has been moved to Supplementary Fig. 1a

3. Fig2 western blot using anti-CT antibody does not include Δ ctx strain as a negative control, it is hard to determine specificity of this antibody. Also, there is no time mention in which samples were collected for this analysis.

Response:

According to your suggestion, Δ ctxAB strain were constructed. Meanwhile, Western blotting were repeated. The results showed no obvious bands for the cholera toxin were evident in Δ ctxAB strain. This suggested that this antibody is specific of cholera toxin. The relative information has now been added in Fig. 4d in revised MS.

For western blotting experiments, the bacteria were grown overnight and diluted 1:100 in fresh AKI medium. When the cultures were grown for 4 h anaerobically and reached an optical density of 0.2, HD-5 was added at a concentration of 50 μ g/mL, followed by 2-2.5h of shaking until reach an optical density of 0.9-1.0
This information has now been added to the Methods section of the revised MS (lines 413-415 of the revised MS).

4. Fig2 qRT-PCR analysis shows increased levels of all 4 transcripts (tcpP, tcpA, toxT, toxR) including toxR. toxR is expressed from an operon that is not associated with either the VPI or CTX elements and not regulated by ToxT or TcpPH.

Response:

Thank you for your comments. Indeed, we made a mistake in Fig. 2b in original MS. When generating the Fig. 2b in original MS, we forgot to change the gene names in primary templates. After careful examination, we found that the *toxR* and *tcpA* in Fig. 2b in original MS should be *tcpA* and *ctxA*. we have fixed the figure accordingly in Fig. 2b in revised MS. Furthermore, we have performed additional qRT-PCR

experiment to analysis the expression of *toxR*. The results showed that expression of *toxR* exhibited no significant change in the presence of HD-5. RNA-seq results also showed that *toxR* was up-regulated only 1.2-fold in the presence of HD-5. This indicated that HD-5 induced TCP operon up-regulation is mediated by *tcpP* rather than *toxR*. This information has now been added to the Results and Discussion sections of the revised MS (lines 123-124 of the revised MS).

5. Fig2 growth of *Vibrio Cholerae*: CFU would be good indication of viability in supplement.

Response:

Thank you for your comments. The corresponding experiment has been performed. To determine the growth curve of each strain, overnight cultures were diluted 1:1000 in a flask containing 200 ml of LB broth with or without HD-5 and incubated at 37 °C with shaking at 180 rpm. A 100 µl aliquot was removed from the flask and suitable dilutions were plated on LB agar plates. The growth curve was determined by cell counts and is expressed in log₁₀ CFU/ml. Experiments were independently performed three times (lines 385-391 in revised MS).

The results showed that bacterial growth remained unaffected with or without 50 µg/mL HD-5 (Fig. 2e in revised MS), which suggests that HD-5-dependent virulence regulation is not due to a metabolic defect caused by the absence of HD-5.

6. Fig3 is overcrowded and can be split into more figures or some of info can be put into supplemented material. Western blot does not contain negative control (e.g. Δ ctx strain). Using very light band of RNAP as a loading control does not serve it purpose and different, well-expressed protein can serve a better, more reliable control.

Response:

Thank you for your comments. Fig. 3 in the original MS has been split into two figures, Fig. 3 and Fig. 4 in the revised MS.

The results showed no obvious bands for the cholera toxin were evident in Δ ctxAB strain. This suggested that this antibody is specific of cholera toxin. The relative

information has now been added in Fig. 4d in revised MS.

RNAP (RNA polymerase) has been widely used as a loading control in Western blotting analysis (*Infect Immun.* 2015 Sep;83(9):3381-95; *Proc Natl Acad Sci U S A.* 2010 Dec 7;107(49):21128-33; *Microbiology.* 2017 Dec;163(12):1902-1911.). The relatively weak band of RNAP observed in blot image may be due to the high expression of cholera toxin. However, RNAP expression levels were remarkably consistent across different experimental conditions, confirming that RNAP is reliable and reproducible as a loading control.

7. Fig4 overcrowded and some of the data can be put into supplement. In vivo competition graph should include Δ rstA as well, so reader would not need to rely on Fig 3h. It is hard to establish, if Δ rstA and Δ tcpP and Δ rstA Δ tcpP mutants are having save colonization defect.

Thank you for your comments. (1). Fig. 4a legend (original MS) has been merged Fig. 4b in revised MS. (2). Fig. 4f, g (original MS) have been moved to Supplementary Fig. 2a-d in revised MS. (3). Original sequence peaks according to the ChIP-seq analyses in Fig. 4d-g (original MS) have been moved to Supplementary Fig. 2e in revised MS.

According to your suggestion, we have now constructed a double mutant Δ carR lacZ- as parent strain and performed additional experiments to show the CI value of Δ carR Δ tcpP versus Δ carR lacZ- was 1.09 and 0.285 in luminal contents and the tissue-associated contents of mice, which was similar to the CI value (1.22 and 0.327) of Δ tcpP versus Δ carR lacZ- in the luminal contents and the tissue-associated contents (Fig. 5h of the revised MS). These results suggest that the influence of TcpP on the colonization of *V. cholerae* is mediated by CarR. These contents have now been added to the Results section of the revised MS (lines 209-216 of the revised MS).

8. At Fig 4; d-g, EMSA experiments: Wild type RstA with His6 on C terminal protein was incubated with 4 different pieces of labeled DNA to determine direct binding of wt-6His protein to different promoters in concentration from 0-800nM. Since RstA needs to be phosphorylated in order to bind DNA, not too much conclusion can be made from this experiment. In order to properly execute this experiment,

phosphorylated RstA or wt RstA with a point mutation mimicking phosphorylated form needs to be used. To determine specificity, competition with “cold” DNA needs to be used as a control. It is VERY unusual to see to use 800nM of transcriptional regulatory protein in EMSA and not to see complete gel shift.

Response:

Thank you for your comments. We have performed electrophoretic mobility shift assays (EMSA) and competition assays. The results showed that at increasing concentrations of phosphorylated CarR (RstA in original MS) protein, slow migrating bands were observed for the FAM-labeled promoters of *tcpP* and *almE* (positive control). Moreover, addition of unlabeled promoters effectively competed for CarR binding to the labeled promoters, and the retarded band disappeared in the presence of 100-fold excess unlabeled promoter DNA (Fig. 5d and Supplementary Fig. 2a). These results indicate that phosphorylated CarR binds specifically to the promoter regions of *tcpP* and *almE* in vitro. In addition, the binding capacity of non-phosphorylated CarR to the promoter regions of *tcpP* and *almE* was significantly reduced compared with that of phosphorylated CarR (Fig. 5e and Supplementary Fig. 2b). Meanwhile, phosphorylated CarR did not bind to the negative control (the promoter regions of *rpoS* and *toxT*) under the same experimental conditions (Supplementary Fig. 2c, d). The relevant information is now described in the Results of the revised manuscript (pg.7-8, lines 184-196).

Legend comments:

1. Western blotting – line 308: which gel was used? 4-20% or 12% SDS PAGE?

Response:

We apologize for this mistake. 4-12% gel was used for Western blotting. The relative information has been corrected in Methods section (lines 420 in revised MS).

2. In conclusion – introduction of current model of regulation of TCP via direct binding of RstA to *tcpP* promoter in HD-5 dependent manner needs to close few technical holes (stat analysis of RNA seq exp, inclusion of negative controls in western blots, inclusion of “cold” (competitive) DNA in EMSA experiments, using activated form of RstA in EMSA, establishing protein binding sequence.

Response:

Thank you for your comments. According to your suggestions, we have now re-analyzed the RNA-seq data (Fig. 1a and Supplementary Data 1 in revised MS), added *ΔctxAB* strain as a negative control in western blotting experiments (Fig. 4d in revised MS), re-done the EMSA and competition assays (Fig. 5d-g and Supplementary Fig. 2a-d in revised MS) and performed the DNaseI footprinting experiments (Fig. 3c in revised MS) to establish CarR binding sequence in *tcpP* promoter in revised MS.

3. Article would benefit from different structure. If authors would choose to still focus on TCP regulation, more attention should be paid to structure of *tcpP* promoter, position of binding of RstA to reference of transcriptional start of *tcpP*, *tcpI* or small RNA *tarA* and overlapping binding site with different transcriptional activators/repressors.

Response:

To identify the CarR binding site in the promoter region of *tcpP*, we performed a dye-based DNase I foot-printing assay and sequence analysis. The results showed that a potential RstA box (5-TAATGAGAATTACT-3) from -186 to -172bp is crucial to the binding ability of CarR to P_{tcpP} . The relevant information is now described in Results, Discussion and Methods section of the revised MS (lines 197-207, 312-317, 494-505 of the revised MS). Previous results showed that AphA, AphB, HapR and CRP can directly bind to *tcpP* promoter (Front Microbiol. 2020 Apr 17; 11:709.; Mol Microbiol. 2001 Jul;41(2):393-407.). In addition, a small RNA (~91 nt), which was named TarA, is located in *tcpP* promoter (Mol Microbiol. 2010 Dec;78(5):1171-81.). However, the CarR binding site in *tcpP* promoter dose not overlap with the binding sequence of AphA, AphB, HapR, CRP and coding sequence of *tarA* (Supplementary Fig. 3a). The relevant information is now described in Discussion of the revised MS (lines 317-323).

4. I would strongly recommend to strengthen paper by including more experiments, reanalyzing data, restructuring the flow, and rewriting discussion. Refining the model with proven data could shed a new light into colonization and virulence of vibrio cholerae and be beneficial for the field.

Response:

We modified the manuscript in the following ways, as you requested. (1). we have added the DNase I footprinting assay, redone the EMSA assay, added $\Delta ctxAB$ strain as negative control in western blotting, added western blotting to analyze CarR level and phosphorylated state of CarR. (2). We have reanalyzed RNA-seq data, reprogrammed Fig. 3 in original MS and Refined Fig. 7 in revised MS. (3). We have rewritten Discussion section according your suggestions. We hope reviewer agree that these revisions significantly enhance the strength and impact of the work.

Reviewer #3 (Remarks to the Author):

Liu et al investigate the response of the cholera pathogen to the human enteric alpha-defensin 5 and find that this peptide promotes *V. cholerae* virulence gene expression by triggering the pathogen's RstAB two component system. They provide evidence that RstA directly activates virulence gene expression by binding to the promoter of the virulence regulator TcpP. In general, the data is convincing, and it provides new knowledge of a molecular pathway by which host factors may trigger the pathogen's virulence program.

The observations that RstA promotes virulence gene expression is new but the introduction of this paper does not place this data within the context of the literature. Reference 37, mentioned here only in the discussion, discovered Vc 1319-20 at the same time as ref 23 and named the RstAB two component system VprAB and showed that its expression was increased by the cationic peptide polymyxin (a similar agent as HD-5). Moreover, both ref 23 and 37 showed that an RstA mutant has a colonization defect. The current paper raises the possibility the colonization defect is due to diminished virulence gene expression, rather than defective lipid A modification, but this idea is not discussed. This may be true because Paneth cells are not mature in 5-day old mice and there is little production of antimicrobial peptides at this point. Thus, it is not possible to use this model to understand *V. cholerae* response to CAMPs in vivo. In any case, their conclusion in the abstract that 'this study established the colonization site recognition mechanism of *V. cholerae*' is an extreme over-statement and should be toned considerably or deleted.

Response:

Thank you for your patient and thoughtful reading as well as the constructive comments and advices about our manuscript. We have revised the manuscript based on your comments and suggestions.

Detailed point-by-point responses are provided below.

Re: “The observations that RstA promotes virulence gene expression is new but the introduction of this paper does not place this data within the context of the literature. Reference 37, mentioned here only in the discussion, discovered Vc 1319-20 at the same time as ref 23 and named the RstAB two component system VprAB and showed that its expression was increased by the cationic peptide polymyxin (a similar agent as HD-5).”

Response:

Thank you for your comments. In *V. cholerae* C6706, CarR has confirmed to contribute to intestinal colonization through endotoxin modification. Furthermore, it has reported that CarSR is involved in virulence gene regulation in various pathogenic bacteria. Meanwhile, the expression of *carR* was increased by the polymyxin B, which is a kind of CAMP. So, we hypothesis that *V. cholerae* may sense HD-5 in small intestine for successful colonization. The relative information has been added in Introduction of revised MS. (lines 81-83 in revised MS)

Re: “Moreover, both ref 23 and 37 showed that an RstA mutant has a colonization defect. The current paper raises the possibility the colonization defect is due to diminished virulence gene expression, rather than defective lipid A modification, but this idea is not discussed. This may be true because Paneth cells are not mature in 5-day old mice and there is little production of antimicrobial peptides at this point. Thus, it is not possible to use this model to understand *V. cholerae* response to CAMPs in vivo”.

Response:

Thank you for your comments. Ref 23 and 37 in original MS showed that, in *V.*

cholerae, CarSR was identified to sense polymyxin B and directly regulate the expression of *almEFG* to resist killing by polymyxin B. However, the Δ *almEFG* did not exhibit defects the colonization capacity of *V. cholerae* and the contribution of CarR to colonization differs between the *V. cholerae* strains. In this study, we revealed that, in *V. cholera* strain EL2382, CarSR promotes virulence gene expression, adhesion in vitro and intestinal colonization in vivo. So, we speculated that *V. cholerae* sense CAMPs in small intestine to activate CarSR, which would further promote the expression of *tcpP* and downstream virulence genes to enhance colonization. The relative information has been added in Discussion of revised MS. (lines 278-288 in revised MS)

Re: “In any case, their conclusion in the abstract that ‘this study established the colonization site recognition mechanism of *V. cholerae*’ is an extreme over-statement and should be toned considerably or deleted.”.

Response:

The sentence “In conclusion, this study established the colonization site recognition mechanism of *V. cholerae*, which plays a critical role in pathogenesis.” in the original manuscript (lines 32-34) has been modified as “In conclusion, this study reveals a complete virulence-regulating pathway, in which the CarSR two-component regulatory system senses HD-5 to activate virulence gene expression in *V. cholera*.” in the revised manuscript (lines 32-35).

General comments

1. RstAB is the name of genes in the cholera toxin prophage in *V. cholerae* and it would be preferable if the name CarSR was used instead.

Response:

Thank you for your comments. “RstAB” has now been changed to “CarSR” in revised MS.

2. Figure 1:

A. Showing the RNA-seq data on circos plots is not informative; generally MA plots

are used to compare transcriptional profiles.

Response:

Thank you for your comments. The circos plots has now been changed to MA plots in revised MS (Fig. 1a).

B. Are the COG categories listed significantly enriched or depleted?

Response:

Thank you for your comments. Differentially expressed genes were grouped into COG categories based on National Center for Biotechnology Information classifications. The significant enrich of a given COG in the sets of up- or downregulated genes were determine using one-tailed Fisher's exact test with Benjamini–Hochberg false discovery ($P < 0.05$). COG categories that are significantly enriched have now been indicated by asterisks in Supplementary Fig. 1a in revised MS.

The COG categories that were significantly enriched in the set of upregulated genes were primarily involved in defense mechanisms, intracellular trafficking, secretion and vesicular transport, signal transduction mechanisms, amino acid transport and metabolism, lipid transport and metabolism, transcription, replication, recombination and repair, cell wall/membrane/envelope biogenesis, posttranslational modification, protein turnover, chaperones and inorganic ion transport, and metabolism (Supplementary Fig. 1a). The COG categories that were significantly enriched in the set of downregulated genes included translation, ribosomal structure and biogenesis, and nucleotide transport and metabolism (Supplementary Fig. 1a).

C. The Fig1 legend does not correspond to the panels in the figure; e.g. legend 1c describes panel a.

Response:

Thank you for pointing it out. The Fig1 legend has been revised in the revised manuscript.

3. Fig 2 and 3, the use of the abbreviation of CT for control (because CT often means cholera toxin) is confusing spell out 'control' or use 0 HD-5.

Response:

Thank you for pointing it out. "CT" has been revised to "Control" throughout the manuscript.

4. Fig 3 and 4, For the CI assays note should be made in the legends that competitions are vs a lacZ- WT strain

Response:

Thank you for your comments. In Fig. 4a of revised MS (Fig. 3e of original MS), CI is defined as the output ratio of mutant strains to WT lacZ- divided by the input ratio of mutant strains to WT lacZ-. In Fig. 5h of revised MS (Fig. 4h of original MS), CI is defined as the output ratio of mutant strains to $\Delta carR$ lacZ- divided by the input ratio of mutant strains to $\Delta carR$ lacZ-. This has been added in the figure legends of Fig. 4a and 5h in the revised MS.

5. Figure 4

A. It's hard to see the shifted band for almE.

Response:

The EMSA experiments were completely redone. We have performed electrophoretic mobility shift assays (EMSA) and competition assays. The results showed that at increasing concentrations of phosphorylated CarR (RstA in original MS) protein, slow migrating bands were observed for the FAM-labeled promoters of *tcpP* and *almE* (positive control). Moreover, addition of unlabeled promoters effectively competed for CarR binding to the labeled promoters, and the retarded band disappeared in the presence of 100-fold excess unlabeled promoter DNA. These results indicate that phosphorylated CarR binds specifically to the promoter regions of *tcpP* and *almE* in vitro. The relative information was shown in Supplementary Fig. 2a-b in revised MS.

B. Note that binding of RstA to the tcpH promoter does not prove that RstA activates

tcpH transcription.

Response:

We are sorry for the inaccurate description. The statement of ‘These results suggest that the regulation of virulence gene expression by RstA in *V. cholerae* is mediated by TcpP.’ has been changed to ‘These results indicate that the influence of TcpP on the colonization of *V. cholerae* is mediated by CarR.’ in revised MS (lines 215-216).

6. Line 205-206 Caco-2 adherence is not equivalent to ‘colonization’.

Response:

‘colonization’ has been changed to adherence’ in revised MS (lines 242).

7. Line 228-229 the claim that they show that *V. cholerae* senses HD-5 in the crypt to confer a colonization advantage in the small intestine’ is an unwarranted. There is no data shown here to show that d5 mice express defensins.

Response:

Thank you for your comments. We are sorry for the confusion of the relevant statement in the original MS. In the revised MS, the content has been changed to “In this study, we revealed that, in *V. cholera* strain EL2382, CarSR promotes virulence gene expression, adhesion in vitro and intestinal colonization in vivo. So, we speculated that *V. cholerae* sense CAMPs in small intestine to activate CarSR, which would further promote the expression of tcpP and downstream virulence genes to confer colonization advantages” (Lines 282-286 of the revised MS).

8. Ref 27 shows that defective colonization in a CarR (RstA) mutant is strain dependent; the authors should mention this fact in the discussion.

Response:

In *V. cholerae*, CarSR was identified to sense polymyxin B and directly regulate the expression of *almEFG* to resist killing by polymyxin B. However, the Δ *almEFG* did not exhibit defects the colonization capacity of *V. cholerae* and the contribution of CarR to colonization differs between the *V. cholerae* strains. In this study, we revealed that,

in *V. cholera* strain EL2382, CarSR promotes virulence gene expression, adhesion in vitro and intestinal colonization in vivo. So, we speculated that *V. cholerae* sense CAMPs in small intestine to activate CarSR, which would further promote the expression of *tcpP* and downstream virulence genes to enhance colonization. Given that CarSR regulation pathway is strain specific, whether this phenomenon also exists in other *V. cholerae* strains need further investigation. The relative information has been added in revised MS. (lines 278-288 in revised MS)

9. In general, the English is understandable but the manuscript still needs some polishing.

Response:

Thank you for your comments. The manuscript has been polished by an English language editing company.

Minor points

1. The methods should state the origin of the *V. cholerae* strain e.g. year of isolation and biotype.

Response:

The *V. cholerae* O1 El Tor strain E12382 isolated in 1994, was kindly provided by Shanghai Municipal Center for Disease Control & Prevention. The relative information has been added in Method section in revised MS (lines 353-354).

2. The methods should state the sources of the HD-5, HD-6, HBD2 and LL-37.

Response:

HD-5 (SP-ADF5-1) and HD-6 (SP-ADF6-5) were purchased from Innovagen (Sweden). HBD2(HY-P7135) and LL-37(HY-P1222) were purchased from MCE (China). The relative information has been added in Method section in revised MS (lines 362-363).

REVIEWERS' COMMENTS:

Reviewer #1 (Remarks to the Author):

The authors have submitted a revised manuscript outlining the role of the RstAB (now named CarRS) two-component regulatory system in the pathogenesis of *Vibrio cholerae*. The authors have addressed many of the comments raised by the reviewers and the resulting manuscript is much improved. I have only minor typographical comments and suggestions for improvement.

Line 209-210 - This is a little confusing - I think the author's mean to say that "Given that CarR regulates tcpP expression, we wanted to determine if the fitness defect of the Δ carR mutant observed in Fig. 4a was due to this effect alone. We therefore constructed a Δ tcpP mutant and a Δ carR Δ tcpP double mutant in *Vibrio cholerae* and competed them against a Δ carR-lacZ mutant alone"

Line 233 - "increased susceptibility" should be "increased expression"

Line 243-244 - Replace with "Adherence to intestinal epithelial cells can be affected by autoaggregation, therefore we tested the autoaggregation of the WT and Δ carR mutants."

Line 259 "then gaining" replace with "gain"

Line 260 replace "was" with "is"

Line 305 - replace "mechanism" with "role"

Line 318 - remove "Also,"

Line 328 - I might modulate the language here to state, "This suggests that CarR plays multiple roles during *V. cholerae* infection"

Figure 5 - panel B - PrpoS should replace rpoS;

Reviewer #2 (Remarks to the Author):

Authors of the paper "*Vibrio cholerae* senses human enteric α -defensin 5 through a CarSR two-component system to promote bacterial pathogenicity" did remarkable job polishing the paper after 1st reviewing process. My major objection were addressed in short time and I feel that paper is suited for publication with very minor (mostly cosmetic) modifications.

I have left in my text sentence numbering, so it will be easier to find it in the text:

32 turn activates downstream virulence genes to promote *V. cholerae* colonization. In
33 conclusion, this study reveals a complete virulence-regulating pathway, in which the
34 CarSR two-component regulatory system senses HD-5 to activate virulence genes
35 expression in *V. cholerae*.

There is no proof that it is a complete virulence pathway. Even a title of paper is addressing role of CarSR as a modifiers of virulence, not a major players.

61 production and intestinal colonization¹³⁻¹⁵. In addition, CaSR or VprAB in *V. cholerae*
62 is involved in virulence regulation in various pathogenic bacteria¹⁶⁻²².

It is most likely typo, should be CarSR, because CaSR is part of intestinal response, not existing in *V. cholerae*

64 However, the specific regulatory mechanism controlling virulence
65 genes expression in *V. cholerae* remains poorly understood.

Maybe influence of CarSR on virulence is not completely understood, but regulatory mechanism of gene expression in *V. cholerae* is extensively studied.

A double

210 mutant Δ carR lacZ- were performed.

Needs to be re-worded, a double mutant can be constructed, not performed.

Also, a small

319 RNA (~91 nt), which we named TarA, is also located in the tcpP promoter⁴⁹. A

Authors did not name small RNA – it is exact copy and paste from original paper.

This indicated that CarR and other regulators

323 cooperate to regulate tcpP expression.

CarR binding to the promoter does not indicate cooperation with other regulators binding to the same promoter – it could also mean competition. Needs re-wording.

487 EDTA and 5% glycerol, with or without 30 mM acetyl phosphate). For CarR gel 488 mobility shift assays, 30 mM acetyl phosphate was added to the binding buffer for 489 generating phosphorylated, active PhoP.

Direct copy and paste from PhoP phosphorylation paper. CarR is being phosphorylated, not PhoP (I hope)

(a) and in the small

770 intestine of mice (b). Data are presented as mean \pm SD (n = 3)

In Figure3 (b) – did not describe condition (it is in text, but not in the legend)

Reviewer #3 (Remarks to the Author):

A thorough response to the critiques has been provided.

REVIEWERS' COMMENTS:

Reviewer #1 (Remarks to the Author):

The authors have submitted a revised manuscript outlining the role of the RstAB (now named CarRS) two-component regulatory system in the pathogenesis of *Vibrio cholerae*. The authors have addressed many of the comments raised by the reviewers and the resulting manuscript is much improved. I have only minor typographical comments and suggestions for improvement.

Response:

Thank you for your positive assessment of our work. We have now corrected these grammatical and typos errors in revised MS, and improved the grammar of the text according to your suggestions.

1.Line 209-210 - This is a little confusing - I think the author's mean to say that "Given that CarR regulates *tcpP* expression, we wanted to determine if the fitness defect of the $\Delta carR$ mutant observed in Fig. 4a was due to this effect alone. We therefore constructed a $\Delta tcpP$ mutant and a $\Delta carR \Delta tcpP$ double mutant in *Vibrio cholerae* and competed them against a $\Delta carR$ -*lacZ* mutant alone"

Response:

Thank you for your comments. The sentence has been changed to "Given that CarR regulates *tcpP* expression, we wanted to determine if the fitness defect of the $\Delta carR$ observed in Fig. 4a was due to this effect alone. We therefore constructed a $\Delta tcpP$ and a $\Delta carR \Delta tcpP$ double mutant in *V. cholerae* and competed them against a $\Delta carR$ *lacZ*-alone." in revised MS. (Lines 207-210)

2.Line 233 - "increased susceptibility" should be "increased expression"

Response:

Done.

3.Line 243-244 - Replace with "Adherence to intestinal epithelial cells can be affected by autoaggregation, therefore we tested the autoaggregation of the WT and $\Delta carR$ mutants."

Response:

Thank you for your comments. The sentence has been changed to "Adherence to intestinal epithelial cells can be affected by autoaggregation, therefore we tested the autoaggregation of the WT and $\Delta carR$." in revised MS. (Lines 243-244)

4.Line 259 "then gaining" replace with "gain"

Response:

Done.

5.Line 260 replace "was" with "is"

Response:

Done.

6.Line 305 - replace "mechanism" with "role"

Response:

Done.

7.Line 318 - remove "Also,"

Response:

Done.

8.Line 328 - I might modulate the language here to state, "This suggests that CarR plays multiple roles during *V. cholerae* infection"

Response:

The sentence has been changed to “This suggests that CarR plays multiple roles during *V. cholerae* infection.” in revised MS. (Lines 327-328)

9.Figure 5 - panel B - PrpoS should replace rpoS;

Response:

Thank you for your comments. In ChIP-qPCR and EMSA experiments, the DNA fragment was cloned in coding region of *rpoS* as negative control. The relative information has been added in revised MS. (Lines 180-181 and 193)

Reviewer #2 (Remarks to the Author):

Authors of the paper “*Vibrio cholerae* senses human enteric α -defensin 5 through a CarSR two-component system to promote bacterial pathogenicity” did remarkable job polishing the paper after 1st reviewing process. My major objection were addressed in short time and I feel that paper is suited for publication with very minor (mostly cosmetic) modifications.

I have left in my text sentence numbering, so it will be easier to find it in the text:

Response:

Thank you once again for your valuable comments. In the light of your suggestions, we made some final edits to our manuscript. We are glad that you offered affirmation to our work.

1. Line 32 - turn activates downstream virulence genes to promote *V. cholerae* colonization. In conclusion, this study reveals a complete virulence-regulating pathway, in which the CarSR two-component regulatory system senses HD-5 to activate virulence gene expression in *V. cholerae*.

There is no proof that it is a complete virulence pathway. Even a title of paper is addressing role of CarSR as a modifier of virulence, not a major player.

Response:

Thank you for your comments. The sentence has been changed to “In conclusion, this study reveals a virulence-regulating pathway, in which the CarSR two-component regulatory system senses HD-5 to activate virulence genes expression in *V. cholerae*.” in revised MS. (Lines 31-33)

2. Line 61 - production and intestinal colonization¹³⁻¹⁵. In addition, CaSR or VprAB in *V. cholerae* is involved in virulence regulation in various pathogenic bacteria.

It is most likely typo, should be CarSR, because CaSR is part of intestinal response, not existing in *V. cholerae*.

Response:

Done.

3. Line 64 - However, the specific regulatory mechanism controlling virulence genes expression in *V. cholerae* remains poorly understood.

Maybe influence of CarSR on virulence is not completely understood, but regulatory mechanism of gene expression in *V. cholerae* is extensively studied.

Response:

Thank you for your comments. The sentence has been changed to “However, the influence of CarSR on virulence in *V. cholerae* is not completely understood.” in revised MS. (Lines 62-63)

4. Line 210 - A double mutant $\Delta carR lacZ$ - were performed.

Needs to be re-worded, a double mutant can be constructed, not performed.

Response:

Thank you for your comments. The sentence has been changed to “Given that CarR regulates *tcpP* expression, we wanted to determine if the fitness defect of the $\Delta carR$ observed in Fig. 4a was due to this effect alone. We therefore constructed a $\Delta tcpP$ and a $\Delta carR \Delta tcpP$ double mutant in *V. cholerae* and competed them against a $\Delta carR lacZ$ -alone.” in revised MS. (Lines 207-210)

5. Line 319 - Also, a small RNA (~91 nt), which we named TarA, is also located in the *tcpP* promoter⁴⁹.

Authors did not name small RNA – it is exact copy and paste from original paper.

Response:

Thank you for your comments. The sentence has been changed to “A small RNA (~91 nt), which was named TarA, is also located in the *tcpP* promoter.” in revised MS. (Lines 317-318)

6. Line 323- This indicated that CarR and other regulators cooperate to regulate *tcpP* expression.

CarR binding to the promoter does not indicate cooperation with other regulators binding to the same promoter – it could also mean competition. Needs re-wording.

Response:

Thank you for your comments. The sentence has been changed to “This indicated that CarR and other regulators cooperate or compete to regulate *tcpP* expression.” in revised MS. (Lines 321-322)

7. Line 487- EDTA and 5% glycerol, with or without 30 mM acetyl phosphate). For CarR gel shift mobility shift assays, 30 mM acetyl phosphate was added to the binding buffer for generating phosphorylated, active PhoP.

Direct copy and paste from PhoP phosphorylation paper. CarR is being phosphorylated, not PhoP (I hope)

Response:

Done. “PhoP” has been changed to “CarR” in revised MS. (Line 495)

8. Line 770 - (a) and in the small intestine of mice (b). Data are presented as mean \pm SD (n = 3)

In Figure 3 (b) – did not describe condition (it is in text, but not in the legend)

Response:

Thank you for your comments. The sentence has been changed to “qRT-PCR analysis showed a 3.92- and 2.53-fold increase in the expression of *carR* and *casS* in AKI medium supplemented with HD-5 (Fig. 3a). In addition, qRT-PCR assays showed that the expression of *carR* and *casS* in the WT strain colonized in the small intestine exhibited a 10.18- and 3.51-fold increase compared to that in LB medium (Fig. 3b).” in revised MS. (Lines 135-139)

Reviewer #3 (Remarks to the Author):

A thorough response to the critiques has been provided.

Response:

Thank you for your consideration of this work. We appreciate your effort in helping us improve our manuscript.